# Learning to Watermark: A Selective Watermarking Framework for Large Language Models via Multi-Objective Optimization

**Chenrui Wang**[1]  **Junyi Shu**[1]  **Billy Chiu**[2]
**Yu Li**[3]  **Saleh Alharbi**[4]  **Min Zhang**[1]  **Jing Li**[1]✉

[1]Harbin Institute of Technology, Shenzhen, China
[2]Lingnan University, China  [3]Zhejiang University, China  [4]Shaqra University, Saudi Arabia
wangchenray111@gmail.com  jingli.phd@hotmail.com

## Abstract

The rapid development of LLMs has raised concerns about their potential misuse, leading to various watermarking schemes that typically offer high detectability. However, existing watermarking techniques often face trade-off between watermark detectability and generated text quality. In this paper, we introduce Learning to Watermark (LTW), a novel selective watermarking framework that leverages multi-objective optimization to effectively balance these competing goals. LTW features a lightweight network that adaptively decides when to apply the watermark by analyzing sentence embeddings, token entropy, and current watermarking ratio. Training of the network involves two specifically constructed loss functions that guide the model toward Pareto-optimal solutions, thereby harmonizing watermark detectability and text quality. By integrating LTW with two baseline watermarking methods, our experimental evaluations demonstrate that LTW significantly enhances text quality without compromising detectability. Our selective watermarking approach offers a new perspective for designing watermarks for LLMs and a way to preserve high text quality for watermarks. The code is publicly available at: https://github.com/fattyray/learning-to-watermark.

## 1 Introduction

The rapid progress of large language models (LLMs) [26, 40, 45] has brought both remarkable capabilities [4, 14, 27, 41] and serious risks of misuse, including copyright concerns [11, 31], academic misuse [36] and other malicious purposes [23, 24]. Consequently, watermarking techniques have emerged as essential tools for detecting and tracing AI-generated texts.

These watermarking techniques embed signals that are imperceptible to humans yet detectable by watermark detection algorithms. Among these methods, the KGW watermark [12] partitions the vocabulary into "green-list" and "red-list" groups, subtly biasing token selection toward the "green-list". This allows detection of watermarked content via statistical hypothesis testing. However, despite their imperceptibility, these watermarks can degrade the text's semantic coherence, as the systematic bias influences token selection contrary to natural model preferences. Recent attempts to address these quality concerns include the TS-watermark [9], which adaptively adjusts watermark strength and green-list ratios; the NS-watermark [37], which limits the extent of watermarking to improve text quality; and EXP-edit [16], which proposes a distortion-free watermark integrated during the sampling process.

---

✉Corresponding author.

39th Conference on Neural Information Processing Systems (NeurIPS 2025).

However, these methods introduce significant trade-offs. The TS-watermark restricts user flexibility by preventing manual adjustment of watermark parameters when either stronger or weaker watermarking is desired. The NS-watermark significantly prolongs text-generation time, and its focus on maintaining text quality compromises detectability. Under its minimal watermarking constraints, the resulting z-score barely surpasses detection thresholds, leaving it vulnerable to trivial attacks—such as changing just one "green-list" token to a "red-list" token—to evade detection. The EXP-edit method, meanwhile, requires hundreds of inference steps for watermark detection, resulting in impractically long detection times, often minutes for even short texts. In contrast, selective watermarking methods aim to watermark only a carefully chosen subset of tokens during text generation. This selective process relies on reproducible selection criteria at detection, enabling efficient watermark verification exclusively on these marked tokens. One notable selective watermarking approach is SWEET [17], proposed specifically for code generation. However, its reliance on manual determination of entropy thresholds, derived through extensive grid-search analyses across sample datasets with varying watermark strengths and "green-list" ratios, hampers its practical usability in real-world applications.

In this paper, we propose a novel selective watermarking approach. Rather than relying exclusively on entropy as a selection criterion, we introduce a lightweight multilayer perceptron, referred to as the Selector Network. This network leverages the sentence embeddings of previously generated text, current token entropy, and the ratio of watermarked tokens within the generated content to adaptively determine when to apply the watermark. By training the Selector Network through multi-objective optimization—using specifically constructed detectability and quality loss functions and the Multiple Gradient Descent Algorithm [5, 35], we guide it toward Pareto-optimal watermarking decisions. This ensures an effective balance between maintaining high watermark detectability and preserving the quality of generated text. In summary, our contributions are as follows:

- **Analysis**: We find existing selective watermarking method [17] underexplored potentially informative factors that may be used as criterions for selection. We are the first to propose the method of utilizing a trained network to make decisions on whether to selectively apply watermark, unveiling a new perspective of selective watermarking strategies.

- **Method**: We propose LTW, a novel selective watermarking framework that uses a trained lightweight network for selectively watermarking LLMs. We introduce LTW-1 and LTW-0, by applying our selective framework to baseline watermark KGW and Unigram.

- **Evaluation**: We conducted extensive experiments across multiple models, demonstrating the high text quality and detectability of our methods. We surpass previous watermarking methods in text quality, having the least perplexity while without compromising detectability.

## 2  Related Work

**Watermarking Methods.**    The rapid advance of LLM and their potential misuse have prompted the need to watermark LLM-generated text to distinguish them from human-written text. Currently, text watermarking methods can be divided into two types, watermarks for generated texts [1, 2, 28, 32, 34, 38, 39, 42] and watermarks for LLMs. The former modify existing text to produce watermarked text, these methods are usually format-based [2, 28, 32, 34], lexical-based [39, 42] or syntax based [1, 38]. For the latter, which takes place during logits generation or during token sampling, implants the watermark during LLM generation. KGW-based [12] methods modifies the logits according to a green/red list, while sampling methods [3, 16] such as Christ's [3] and EXP-edit [16] watermark by using their pseudo random sampling methods. Recent works of LLM watermarking are proposed to reduce text quality degradation caused by adding watermarks or to increase their robustness under attacks. For example, Unigram [46] uses a fixed random key to produce their "green/red list", while SIR [19] uses semantics to determine the "green-list" to improve robustness under attack. Token-Specific Watermark [9] trains $\gamma$ and $\delta$ generators to alter these hyperparameters during token generation to obtain better semantic coherence. To address the low-entropy limitation that previous works [3, 12, 17] has shown, SWEET [17] provided a selective watermarking method to improve the quality of watermarked codes generated by LLMs, and EWD [22] provided an entropy-based detection method to improve detection ability.

**Balancing Detection and Quality.**    Recent works [8, 9, 18, 21, 37] indicate watermarking brings text quality decay and some works [6, 17] indicate that in some tasks such as coding, watermarking

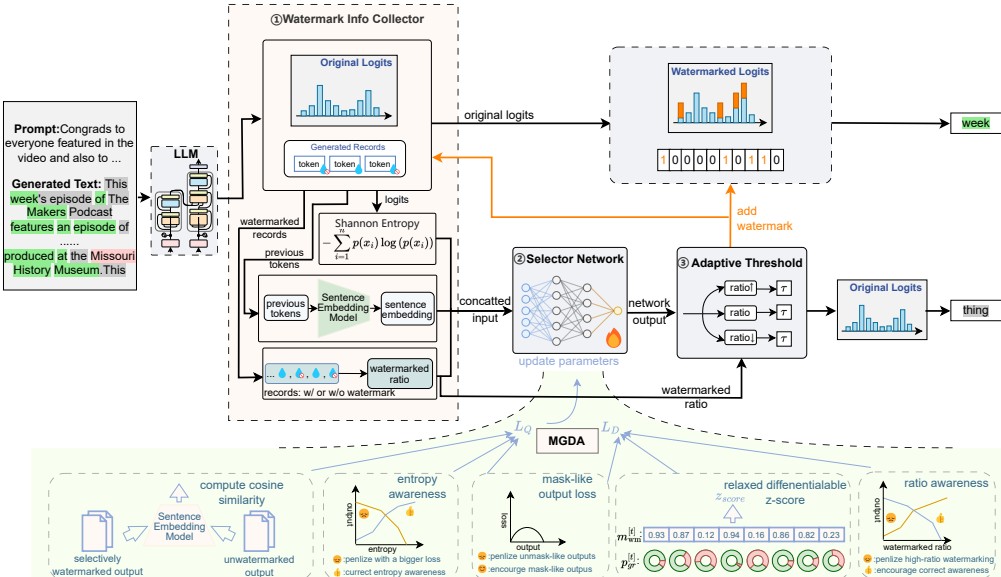

Figure 1: An illustration of our selective watermarking method. (1) The upper part illustrates that at each generation time step, the watermark collector concatenates the calculated input for the trained Selector Network, whose output is compared to an adaptive threshold to decide whether to watermark the current token. (2) The lower part is about constructing the detection oriented and text quality oriented loss, and utilize the MGDA algorithm to tackle this multi-objective optimization problem and optimize towards Pareto optimal.

causes a notable performance decline. It is understandable as some crucial tokens the LLM should generate were changes due to watermarking, resulting in the problem. Methods for solving this include detection trade-off methods [18, 37] and selective watermarking method [17]. NS-watermark [37] indicates that a minimal number of watermarked text is needed to achieve detection, and developed a watermark to meet the constraint; however, the method hurts detection ability because unlike selective watermarking, without the ability to determine which tokens are selected and only detecting them, the method has to detect all generated text while only a fraction of them are watermarked, leaving their watermark with a low z-score and can be easily removed under their minimal constraint. Other detection trade-off method [18] add bias to tokens that are in the "red-list" but are deemed important. SWEET [17] proposed a selective watermarking method based on entropy threshold, which only watermarks and detects high-entropy tokens. Some work [21] has applied the selective watermarking method as a pre-processing step for improving text quality in their watermark.

**Multi-Objective Optimization.** In LLM watermarking, detection and text quality are a set of optimization objectives that often conflict with each other. Previous work [9] addressed this problem by using the Multiple-gradient Descent Algorithm (MGDA) [5, 35] to optimize the two objectives. They trained $\gamma$ and $\delta$ generators, which takes the embedding of the last token as input, to adaptively provide these two hyperparameters for watermarking and detection.

## 3   Methodology

We believe that the selective watermarking approach holds significant promise for simultane-ously achieving high watermark detectability while preserving text quality. However, existing research [17, 21] has focused primarily on using entropy as the sole criterion for selection, leaving other potentially informative factors underexplored. In this work, we followed prior work of selec-tive watermarking [17] and proposed incorporating semantic embedding of the previous text and watermarked ratio into the selection process together with entropy for selective watermarking. We

take them as input for our trained network to determine when to selectively add the watermark. We introduce our novel approach for selective watermarking, LTW.

## 3.1 Problem Formulation

Following the standard approach for watermarking LLMs [12] by modifying the output logits $\mathbf{l}_t \in \mathbb{R}^{|\mathcal{V}|}$ at each generation step $t$, for each token $v$ in the vocabulary $\mathcal{V}$, is assigned to the green-list or red-list according to a pseudorandom generator $R$, seeded with a key $\hat{k}$. $\mathcal{V}$ is shuffled to divide the vocabulary $\mathcal{V}$ into two disjoint subsets : $L_{\text{green}}, L_{\text{red}} = \text{Shuffle}(\mathcal{V}, R(\hat{k}), \gamma)$ where $|L_{\text{green}}| = \gamma |\mathcal{V}|$ and $\gamma \in (0,1)$ is a hyperparameter controlling the green-list ratio and $\delta$ is the hyperparameter controlling the watermark strength. From this partition, the green-list mask $\mathbf{m}_{\text{green}}(L_{\text{green}}, L_{\text{red}}) \in \{0,1\}^{|\mathcal{V}|}$ is denoted as:

$$\mathbf{m}_{\text{green}}[v] = \begin{cases} 1, & v \in L_{\text{green}} \\ 0, & v \in L_{\text{red}} \end{cases} \tag{1}$$

We introduce a lightweight multi-layer perceptron $\mathcal{M}_\theta$, which produces a mask for deciding the appropriate generation time step to add the watermark. To enable gradient backpropagation during training, the network is designed to produce continuous output $m_{wm} \in [0,1]$, rather than discrete binary values such as those from a Bernoulli distribution that are not differentiable. To encourage the output values to be close to either 0 or 1, thereby promoting a mask-like behavior, we introduce a regularization term in the loss function that penalizes uncertainty and drives the predictions towards binary-like extremes. We denote the token sequence at the current timestep as $s = [t_1, t_2, \ldots, t_n]$, $\mathcal{E}_{\text{sem}}$ as a semantic model which produces a sentence embedding according to previous $k$ tokens. Let $e \in \mathbb{R}$ denote the Shannon entropy of the token probability distribution at the current generation step and let $r \in [0,1]$ denote the current proportion of watermarked tokens in the generated sequence.

$$\mathbf{m}_{\text{wm}} = \mathcal{M}_\theta \left( \mathcal{E}_{\text{sem}}(s_{n-k+1:n}), \ e, \ r \right) \in [0,1] \tag{2}$$

Combining the green-list and selective mask, we obtain the final perturbation mask over the logits, and the modified logits are given by:

$$\tilde{\mathbf{l}}_{t+1} = \mathbf{l}_{t+1} + \delta \cdot \mathbf{m}_{\text{wm}}^{[t+1]} m_{\text{green}}(L_{\text{green}}, L_{\text{red}}) \tag{3}$$

The selectively watermarked logits, are then transformed into probability using the softmax function, and used for sampling the next token, the process continues until reaches the maximum number of new tokens or the eos token.

The detection process is formulated as a z-test : $z = \frac{|s|_G - \gamma T_{\text{wm}}}{\sqrt{T_{\text{wm}} \gamma (1-\gamma)}}$. The number of all the selected tokens are denoted as $T_{\text{wm}} = \sum_{t=1}^{T} m_{\text{wm}}^{[t]}$ and $|s|_G = \sum_{t=1}^{T} m_{\text{wm}}^{[t]} \cdot p_{gr}^{[t]}$ . $p_{gr}$ is the probability of selecting a green token, relaxing $|s|_G$ from the total number of green tokens to this makes $z$ differentiable for training.

The equations above are our relaxed version of the standard watermarking procedure [12] to make it consistent with our training process. The continuous and differentiable $\mathbf{m}_{\text{wm}}$ together with a differentiable approximation of $z$ enable gradient-based optimization of the model parameters. During inference, however, $\mathbf{m}_{\text{wm}}$ is thresholded by a predefined hyperparameter $\tau \in (0,1)$ as the following Equation 4 suggests, and $p_{gr}^{[t]}$ is either 0 or 1 depending on whether the sampled token is in the green-list. Thus making our equation consistent with the standard process [12]. Further proof of our method is provided in Appendix A.

$$m_{\text{wm}} = \begin{cases} 1, & \text{if } \mathbf{m}_{\text{wm}} > \tau \\ 0, & \text{otherwise} \end{cases} \tag{4}$$

## 3.2 Structure Design

**Selector Network.** The Selector Network is the module in our work that decides whether the current token should be watermarked or not. The network is a multilayer perception [33] that takes the concatenated information during generation as input. These generation related information includes the sentence embedding of the previous tokens, the entropy of the probability distribution for the token to be predicted, and the proportion of generated tokens that are selected to be watermarked. These information can also be obtained during detection, allowing trace back to those selectively

watermarked tokens to be detected on. We adopt SimCSE [7] to provide the sentence embedding which is used as input by our network. The obtained embeddings are first subjected to dimensionality reduction via the MLP layers, then the reduced representations are concatenated with two additional features and fed into a second MLP for further dimensionality reduction. Finally, the model produces an output $\mathbf{m}_{\text{wm}} \in [0, 1]$. Further discussion of its design is given in Appendix E.

**Watermarking Information Collector.** Compared to existing watermarking methods, our approach requires the collection of more comprehensive information during watermarking. To facilitate this, we implement an information collecting module that maintains contextual data from previously generated tokens. Specifically, at each generation step, the token id or embedding of the newly generated token is appended to this module. and the sentence embedding is constructed based on the last k tokens. Furthermore, the watermarked ratio information is updated according to whether the previous token was watermarked. We also compute the entropy of the probability distribution for the current token in this module. These provide the adequate information and are concatenated as input to be used by our network during training or watermarking.

**Adaptive Threshold.** In Section 3.1, we define Equation 4. During inference, we need to discretize the output $\mathbf{m}_{\text{wm}} \in [0, 1]$ into a binary decision of 0 or 1, indicating whether to watermark the current token being generated or not. A dynamic threshold is applied for this purpose. A lower threshold is used when the current watermarking ratio is low, allowing more tokens to be watermarked and thus improving detectability; Conversely, a higher threshold is adopted when having a sufficiently high watermarked ratio.

### 3.3 Learning to watermark: Training for Selective Watermark Insertion

Our goal is to train a selective watermarking module capable of making informed and appropriate watermark insertion decisions. The output of the selective watermarking module is continuous rather than a discrete binary value, serving as a differentiable signal that enables training through gradient-based optimization. Our objective is to achieve high watermark detectability and high generation quality through selective watermarking, which naturally formulates as a multi-objective optimization (MOO) problem. To address this, we adopt the Multiple Gradient Descent Algorithm (MGDA) [5, 35] used by previous work [9] to effectively handle these competing objectives. Moreover, we expect the module to exhibit the following properties:

- First, the output should exhibit mask-like behavior, encouraging the module to make sharp, near-binary decisions regarding whether to embed a watermark.

- Second, the module should be entropy-aware. It should prefer watermarking when the entropy over the probability distribution of the next token is high, and avoid watermarking when it is low. This design is motivated by prior works [17, 22], which have demonstrated that applying watermarks to low-entropy tokens can lead to degraded text quality and reduced watermark detectability.

- Third, the module should provide adaptive control based on the current watermarking ratio. Specifically, when the overall proportion of watermarked tokens is low, the module should be more inclined to add watermarks to maintain detectability. Conversely, when the ratio is already high, it should reduce watermarking to better preserve text quality.

These goals and behavioral characteristics can be formulated as two distinct optimization objectives:

**Preserving Quality.** Part of this is trying to maximize the semantic similarity between the marked text and the unmarked one, aiming to minimize the quality drop brought by adding watermarks. This is achieved by computing the cosine similarity between the sentence embedding of the watermarked output embedding and the non-watermarked one. $E_w$ and $E_s$ denote the semantic embedding in the following loss function 5. The objective also incorporates entropy-aware behavior: the module is encouraged to apply watermarks to high-entropy tokens while avoid applying those with low-entropy, we formulate the following loss function in Equation 6 to describe this objective. In which $\lambda_{\text{e}}$ and $\mu_{\text{e}}$ are a set of hyperparameters, and $e = -\sum_{i=1}^{n} p_i \log p_i$ represents the current entropy.

$$\mathcal{L}_S = -\cos_{\text{sim}}(E_w, E_s) \tag{5}$$

$$\mathcal{L}_{\text{entropy}} = \text{BCE}(m_{\text{wm}}, \sigma(\lambda_{\text{e}}(e - \mu_{\text{e}}))) \tag{6}$$

**Maximizing Detectability.** Second, we seek to maximize watermark detectability, which involves increasing the overall z-score used for detection $z = \frac{|s|_G - \gamma T_{\text{wm}}}{\sqrt{T_{\text{wm}}\gamma(1-\gamma)}}$. To make the z-score differentiable, instead of the sum of the green tokens among those chosen to be watermarked, it is relaxed to the probability of producing a green token among those chosen. This can be denoted as $|s|_G = \sum_{t=1}^{T} p_{gr}^{[t]} \cdot m_{\text{wm}}^{[t]}$ forming the following Equation 7. Furthermore, this objective promotes adaptive watermarking behavior: when the overall watermarking ratio is low, the model is encouraged to watermark for ensuring detectability; While when the ratio is high, it prefers to watermark less, thereby preserving generation quality when detectability can be ensured, we denote $r_t$ as the current watermarked ratio, $f : (0,1) \rightarrow (0,1)$ as a monotonically decreasing function, we construct the following objective as the loss function in Equation 8.

$$z = \frac{\sum_{t=1}^{T} p_{gr}^{[t]} \cdot m_{\text{wm}}^{[t]} - \gamma \sum_{t=1}^{T} m_{\text{wm}}^{[t]}}{\sqrt{\sum_{t=1}^{T} m_{\text{wm}}^{[t]}\gamma(1-\gamma)}} \tag{7}$$

$$\mathcal{L}_{wm\_ratio} = \text{MSE}\left(m_{wm}^{[t]}, f(r_t)\right) \tag{8}$$

The following loss function in Equation 9 describes the need for a mask-like output, penalizing those unmask-like outputs. It is applied to both the following objectives to make the output mask-like.

$$\mathcal{L}_{output\_fix} = -\frac{1}{T}\sum_{t=1}^{T}\left(m_{wm}^{[t]} - 0.5\right)^2 \tag{9}$$

These objectives are jointly formulated as the loss functions in Equation 10 and 11. To achieve a Pareto-optimal solution between quality oriented $L_Q$ and detectability oriented $L_D$, the MGDA [5, 35] algorithm is adopted to determine a descent direction towards it. Further details are provided in the Appendix G.

$$\mathcal{L}_Q = \lambda_{sim}\mathcal{L}_S + \lambda_{entropy}\mathcal{L}_{entropy} + \lambda_{fix}\mathcal{L}_{output\_fix} \tag{10}$$

$$\mathcal{L}_D = -\lambda_z z + \lambda_{wm}\mathcal{L}_{wm\_ratio} + \lambda_{fix}\mathcal{L}_{output\_fix} \tag{11}$$

where $\mathcal{L}_S$ denotes the similarity loss presented in Equation 5, $\mathcal{L}_{entropy}$ denoted in Equation 6 is a loss for achieving entropy awareness, $\lambda_{sim}$ and $\lambda_{entropy}$ are the weighting hyperparameter for the two loss functions. In Equation 11, $z$ denotes our differentiable z-score described in Equation 7, $\mathcal{L}_{wm\_ratio}$ presented in Equation 8 is the loss for incorporating ratio awareness behavior, $\lambda_z$ and $\lambda_{wm}$ are the weighting hyperparameter for them. The $\mathcal{L}_{output\_fix}$ in both functions is for promoting mask-like outputs, $\lambda_{fix}$ is its weighting hyperparameter.

## 4 Experiments

### 4.1 Experiment Setting

**Datasets and Models.** We trained and tested our method on the RealNewsLike subset of the C4 dataset [29], following previous watermark works [8, 9, 12, 13, 21]. We used 10,000 texts for training, and randomly divided another 500 texts as test set. We trained our selective watermarking module using OPT-1.3b [45]. Further discussion about training are provided in Appendix F.1. We then evaluated the performance of our watermark and other watermark methods using OPT-6.7b and GPT-J-6B [40]. After which, we conducted paraphrase attack on the watermarked texts using the Dipper model [15], we followed previous work's [13] parameter settings, using a lex diversity of 60.

**Baselines.** We applied our selective watermarking method to KGW and Unigram to evaluate the effectiveness of our watermarking method, we will denote them as LTW-1 and LTW-0 in the following text, the procedure of our selective method can be found in Appendix C. We compared our methods with KGW [12], Unigram [46], EXP-edit [16], SWEET [17] and Token-Specific [9] under fixed hyperparameters settings. To further analyze the trade-offs between detectability and text quality, as well as between robustness and text quality, we varied the watermark strength $\delta$, generated watermarked texts using our methods and KGW for each strength, and compared their performance by fitting a curve for comparing the trade-off between TPR and perplexity. Further discussion on hyperparameters settings and experiment details are provided in Appendix F.2.

Table 1: Performance comparison of different watermarking methods on OPT-6.7B in terms of Detectability, Robustness, and Text Quality. Higher performance when not under attack indicates higher detectability, higher performance under paraphrase attack shows the robustness of watermark method. Lower perplexity and higher similarity with reference text indicates higher text quality.

| Method | No Attack | | | Dipper Attack | | | Text Quality | |
|--------|-----------|----------|----------|---------------|----------|----------|--------------|------------|
| | TPR@2 | AUROC | Best F1 | TPR@10 | AUROC | Best F1 | Perplexity | Similarity |
| KGW | 0.998 | 0.99990 | 0.99900 | 0.910 | 0.96987 | 0.91465 | 20.5234 | 0.5472 |
| Unigram | 1.000 | 1.00000 | 1.00000 | **0.962** | 0.98217 | **0.95238** | 17.6761 | 0.5313 |
| EXP-edit | 0.972 | 0.98742 | 0.98008 | 0.790 | 0.90527 | 0.83871 | 23.9298 | 0.5046 |
| SWEET | 1.000 | 1.00000 | 1.00000 | 0.924 | 0.96955 | 0.92245 | 17.9301 | 0.5554 |
| TS-watermark | 0.996 | 0.99973 | 0.99598 | 0.810 | 0.93179 | 0.85076 | 14.0804 | 0.5645 |
| LTW-1 (ours) | **1.000** | **1.00000** | **1.00000** | 0.852 | 0.94034 | 0.87810 | **13.6205** | 0.5701 |
| LTW-0 (ours) | **1.000** | **1.00000** | **1.00000** | 0.954 | **0.98326** | 0.94190 | 13.6238 | **0.5735** |

Table 2: Performance comparison of different watermark methods on GPT-J-6B.

| Method | No Attack | | Dipper Attack | | | Text Quality | |
|--------|-----------|----------|---------------|----------|----------|--------------|------------|
| | AUROC | Best F1 | TPR@10 | AUROC | Best F1 | Perplexity | Similarity |
| KGW | 1.00000 | 1.00000 | 0.800 | 0.93668 | 0.87125 | 16.5485 | 0.4941 |
| Unigram | 1.00000 | 1.00000 | **0.944** | **0.97938** | **0.94586** | 13.0226 | 0.4634 |
| EXP-edit | 0.99996 | 0.99701 | 0.860 | 0.93651 | 0.88377 | 21.8741 | 0.4498 |
| SWEET | 1.00000 | 1.00000 | 0.878 | 0.94406 | 0.88777 | 15.2203 | 0.5049 |
| TS-watermark | 1.00000 | 1.00000 | 0.822 | 0.93150 | 0.86068 | 11.7829 | 0.5220 |
| LTW-1 (ours) | **1.00000** | **1.00000** | 0.858 | 0.94813 | 0.88303 | 11.6877 | **0.5286** |
| LTW-0 (ours) | **1.00000** | **1.00000** | 0.926 | 0.96238 | 0.91952 | **9.5571** | 0.4822 |

**Evaluation Metrics.** To evaluate watermark detection performance and robustness against paraphrasing attacks, we employed multiple evaluation metrics: AUROC, Best F1 score, and True Positive Rate (TPR) at specified False Positive Rate (FPR) thresholds. We compared the quality of the watermarked text using two evaluation metrics: Perplexity and cosine similarity between the sentence embeddings of generated text and the reference text. For each text in the test set, we designate the 200 words following the prompt segment as human-written reference text. Sentence embeddings are obtained using the Sentence-BERT model [30].

## 4.2 Experimental Results

Table 1 and Table 2 show results of our watermark method and baseline methods in terms of detectability when not under attack, robustness when under paraphrase attack as well as the text quality of the watermarked text generated using OPT-6.7B and GPT-J-6B. Figure 2, 3 and 4 present a detailed comparison of text quality in the main experiment described above. Figure 5 and 6 show the results of our method with KGW under different watermark strengths. An example of our watermark can be found in the Appendix D, in which compared with baseline methods, our method not only improved text quality, but also improved detection ability.

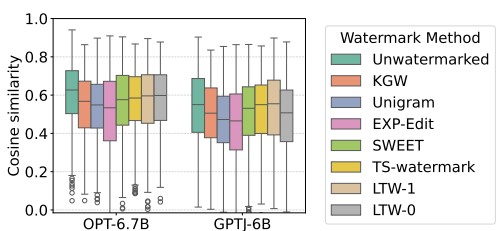

Figure 2: Comparison of the Cosine Similarity between reference text and watermarked text across various LLMs.

**Comparison on Text Quality.** In Figure 3 and 4, we compared the perplexity of our watermark method with the unwatermarked text and other baseline methods. A lower perplexity suggests better text quality. Among the watermark methods, our methods have the least perplexity, implying the least quality degradation among the baselines. In Figure 2, We evaluated text quality by computing the cosine similarity between the sentence embeddings of the original text and the generated texts, a higher similarity suggests better quality and semantic coherence. We observe that, overall, our methods have better semantic similarity compared

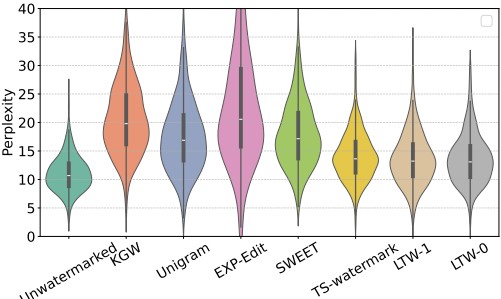

Figure 3: Distribution of perplexity among the unwatermarked text generated using OPT-6.7b and that of texts watermarked using various watermark methods.

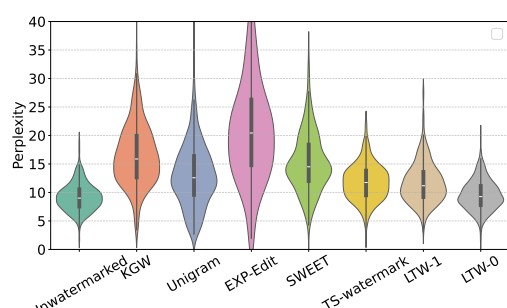

Figure 4: Distribution of perplexity among the unwatermarked text generated using GPT-J-6b and that of texts watermarked using various watermark methods.

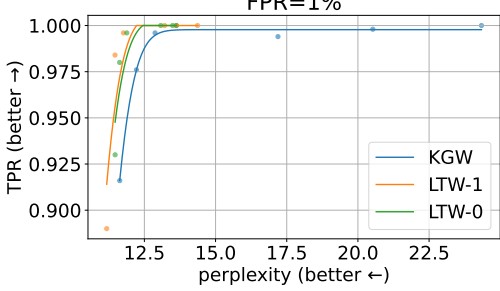

Figure 5: Performance comparison between our watermarking method and KGW watermark in terms of perplexity and TPR across various watermark strengths.

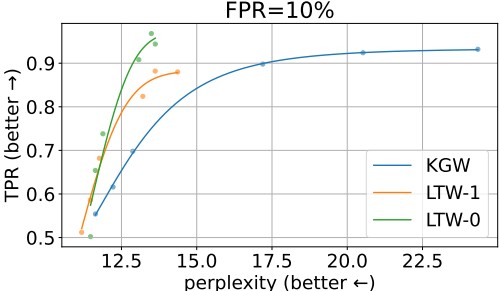

Figure 6: Comparison of our method and KGW under Dipper attack in terms of perplexity and TPR across various watermark strengths.

with other watermarkmethods. Among which, LTW-1 achieved better semantic similarity than all five baseline methods, indicating better semantic coherence and text quality. This performance improvement can be attributed to our selective watermarking mechanism, which learns to insert watermarks based on semantic embeddings and entropy, tending to watermark tokens which carry less semantic significance, while omitting those more critical tokens, thereby minimizing disruption to the overall sentence meaning.

**Comparison on Detectability.** In Table 1 and 2, we compared the detectability of our watermarking methods with baseline methods on OPT-6.7B and GPT-J-6B under a fixed watermark strength. We observe that, although most watermarks do well in detectability, our method consistently demonstrate highly competitive performance across all detection metrics. Furthermore, in Figure 5, we conducted additional experiments comparing the performance between our watermark and KGW across varying watermark strengths by increasing $\delta$. The results show that our approach is clearly superior to KGW, achieving a more favorable Pareto frontier than it, offering improved detectability at the same level of perplexity. Compared to KGW, our methods yield significantly lower perplexity at the same watermark strength and achieves TPR = 1 at a much lower strength.

**Comparison on Robustness.** In Figure 6, we compared the robustness of our method and KGW against paraphrase attack across varying watermark strengths. The results show that our methods can achieve a superior Pareto frontier, showing more robustness than the baseline method KGW. Results on Table 1 and 2 also show that LTW-0 can outperform most baseline methods in robustness while still having lower perplexity. Though method like TS-watermark can achieve a text quality similar to ours, they are less robust when under paraphrase attack.

Table 3: Performance comparison of detectability and quality between using the adaptive threshold module and after removing the adaptive threshold module under different watermark strengths.

| Settings | Metrics | $\delta = 1.5$ | $\delta = 2$ | $\delta = 2.5$ | $\delta = 3$ | $\delta = 3.5$ | $\delta = 4$ |
|---|---|---|---|---|---|---|---|
| **With** | **Perplexity** | 11.686 | 12.354 | 13.003 | 13.410 | 13.981 | 14.029 |
| | **Z-score** | 6.844 | 9.293 | 11.243 | 12.670 | 13.776 | 14.497 |
| **Without** | **Perplexity** | 11.801 | 12.524 | 13.217 | 13.403 | 13.922 | 14.308 |
| | **Z-score** | 6.831 | 9.031 | 11.120 | 12.443 | 13.570 | 14.369 |

## 5 Analysis

**Ablation Studies.** In this section, we focus on assessing the effectiveness of our adaptive threshold module. We conduct ablation studies by removing this module and replacing it with a fixed threshold, and evaluate the perplexity and z-score across varying watermark strengths. The results in Table 3 show that the adaptive threshold can improve detectability in all watermark strength settings, and having a lower perplexity in 4 of 6 watermark strengths settings. Indicating that such a module can help achieve higher detection abilities while having similar or even better text quality.

**Analysis of Network Output.** In this section, we explore the correlation between the output of the trained network and the input. Our network utilize the sentence embedding of previous text as part of its input, though it is hard to find a measurable representation for the embedding, it nevertheless is correlated with the part-of-speech (POS) at the current time step systematically. The results are shown in Figure 7. Our network tends not to watermark adposition, conjunction, punctuations and symbols. Watermarking and altering them may yield semantically incoherent phrases, misinterpreted sentences and wrong equations. Our network more oftenly tends to watermark adverbs and adjectives, which often convey descriptive or qualitative information and have more replacements in the vocabulary. Watermarking them may have a smaller influence on semantic coherence. We also evaluated the correlation between the output and the entropy input at the current time step, which has a positive correlation, encouraging watermarking when having high entropy, detailed discussion are in Appendix D. The results show that both semantic-level features as well as entropy plays an important role in the process of making the selective decision in our method.

**Comparison on Our Two Methods.** Unlike previous work [17] that only applied their selective method to KGW, we applied our selective method to two representative watermarks, KGW and Unigram. In terms of text quality, both watermark variants exhibit significantly lower perplexity and higher semantic similarity after incorporating our selective watermarking method. KGW also shows improved detectability under the OPT model and enhanced robustness against paraphrasing attacks under GPT-J model after applying our selective method. While both of our methods yield similar text quality under the OPT model, they each excel in different aspects under the GPT-J model; Notably, the use of a fixed

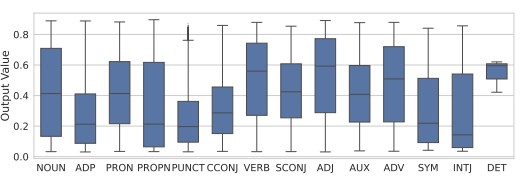

Figure 7: Distribution of the network output across different part-of-speech categories of the token generated at the current time step.

green/red list in the Unigram approach proves beneficial to maintaining robustness under our selective watermarking framework. The enhanced robustness of LTW-0 can be attributed to the use of a fixed green/red list partition, mitigating the risk of detection failure caused by changes in the partition of the red-green list due to changed previous token when under paraphrase attacks. Moreover, selective watermarking alleviates the issue of systematically reduced probability of the important words in the fixed red list, minimizing degradation of text quality and semantic coherence for Unigram.

**Analysis of Measures taken for Robustness.** Applying the hard binary mask instead of keeping the soft weights of the selector network is a design for robustness. For the soft mask we used during training, the mask is also a detection weight as in Equation 7, a token with a small mask is also given light detection weight. We believe that, comparatively, using the hard mask can better enhance the

robustness of the watermarking scheme under paraphrasing attacks. From a design perspective, it is relatively simple to categorize the outputs of the selector into apply watermark and do not apply watermark, and only apply watermark to the first category. During detection, we only need to verify the outputs that were marked for watermarking, which is a simple approach. However, if a soft mask was to be adopted at inference time, the detector would need to consider the soft mask values as per-token detection weights. In such a case, tokens with very low mask values would contribute minimally to detection, while the opposite for those with a high mask value. While this offers finer granularity, it introduces more complexity and uncertainty—especially making it more vulnerable to paraphrasing attacks.

Paraphrasing attacks are hard to change apply watermark to do not apply watermark compared with changing the output values. In our Equation 9, we penalize those unmasklike outputs, making most outputs further from the threshold, thus making it less sensitive to attacks. In contrast, soft mask values are continuous and more sensitive to subtle changes, meaning that a paraphrasing attack may cause small shifts in mask values (e.g., a slightly increased value for a previously low-scored token, or a slightly decreased one for a previously high-scored token). These small perturbations can accumulate and increase the uncertainty of watermark detection, thus reducing robustness.

Choosing a moderate window size for the sentence embedding used by the selector network is also important for robustness. A window size that is too small may cause the sentence embedding to degrade like those of a token-level representation, making it less robust under paraphrasing attacks. This is because paraphrasing typically preserves the overall sentence meaning while altering individual words through synonym substitution or structural changes. At the token level, such variations can affect the selector network's output, making it less robust to these attacks. In contrast, by selecting an appropriately sized window, we can maintain semantic consistency with the paraphrased version, thus improving the robustness of the selector. On the other hand, with a too large window size, new tokens generated will have only a minor influence to the content in the window, which may reduce the sensitivity of the selector to the content being generated.

**Justify for choice of metrics**   We have adopted perplexity and Similarity between the sentence embeddings as our metrics for evaluating Text Quality. Perplexity (PPL) is a natural metric for judging watermark impact because it directly measures the negative log-probability of the generated token sequence. If a watermarking method perturbs the generation more, the generated outputs will typically become less likely to be generated and thus exhibit higher PPL. Generally, lower PPL indicates higher text quality[20]. And it have been used as a metric in previous works [8, 12, 21, 22] as well. When the task has human reference answers such as the task of Text Completion, a higher similarity between the embedding of the reference text and the generated watermarked text captures whether the watermarked text still preserves task-relevant semantics and coherence, remaining appropriate and faithful to the reference text. Previous works [9, 25, 43, 44] have also adopted semantic score in their works and among which, some works [9, 44] also adopted sentence embedding models  [7, 30] for calculating the sentence-level score.

## 6   Conclusions

In this paper, we propose a novel approach for selectively applying watermarks to LLMs, aiming to mitigate the quality degradation caused by watermarking while preserving high detectability. We constructed two distinct loss functions and employed the MGDA algorithm to solve this multi-objective optimization problem. Experimental results demonstrate that our approach achieves superior text quality without compromising detection performance. In summary, our method unveils a new perspective on the design of selective watermarking strategies for LLMs.

## Acknowledgements

This work was supported in part by National Natural Science Foundation of China (62476070), Shenzhen Science and Technology Program (JCYJ20241202123503005, GXWD20231128103232001, ZDSYS20230626091203008, KQTD20240729102154066), Department of Science and Technology of Guangdong (2024A1515011540) and National Key R&D Program of China (SQ2024YFE0200592).

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

# A  Theoretical Justification of LTW as a Selective Watermark

If a watermarking scheme is designed to be selective, it must not rely on random mechanisms or rules that are unverifiable during the detection phase. Otherwise, without a complementary detection strategy capable of identifying which tokens in a generated sequence were chosen to be watermarked during applying watermark, the watermark detection process becomes unreliable.

Specifically, consider the hypothesis testing formulation:

$$z_{\text{orig}} = \frac{|\mathcal{S}|_G - \gamma T}{\sqrt{T\gamma(1-\gamma)}}$$

where $\gamma \in (0,1)$ is the green-list ratio, $T$ is the total number of tokens to be detected, and $|\mathcal{S}|_G$ denotes the number of tokens detected which are in the green-list $G$. Suppose that we define the original test statistic as $z_{\text{orig}}$ corresponding to a total of $T$ tokens, among which $|\mathcal{S}|_G$ are green tokens.

If, during detection, a token that was not selected for watermarking during generation is nonetheless included in the detection process, then the total token count increases to $T+1$, while the expectation of green tokens becomes $|\mathcal{S}|_G + \gamma$.

The updated test statistic becomes:

$$z_{\text{new}} = \frac{(|\mathcal{S}|_G + \gamma) - \gamma(T+1)}{\sqrt{(T+1)\gamma(1-\gamma)}} = \frac{|\mathcal{S}|_G - \gamma T}{\sqrt{(T+1)\gamma(1-\gamma)}}$$

It is evident that $z_{\text{new}} < z_{\text{orig}}$, suggesting that including tokens that were not selected for watermarking into the detection process will, in expectation, degrade the detectability of the watermark.

Therefore, in selective watermarking schemes, it is essential that the selection rule used during watermark embedding be reproducible during detection. Only by reliably identifying the subset of tokens that were eligible for watermarking at generation time can we perform valid and powerful hypothesis testing on that subset alone, ensuring accurate detection outcomes.

Let the generated sequence up to step $n$ be denoted as $s = [t_1, t_2, \ldots, t_n]$. Let $\mathcal{E}$ as the Sentence Embedding model, $\mathcal{V} = \{x_1, x_2, \ldots, x_k\}$ as the vocabulary containing $k$ tokens and $\mathcal{M}_\theta$ as our trained network. Specifically, during the watermark embedding phase, we leverage three key signals:

- 1. The semantic embedding, obtained by passing the preceding $k$ tokens to the sentence embedding model;We denote the obtained sentence embedding as $E = \mathcal{E}_{\text{sem}}(s_{n-k+1:n})$
- 2. The Shannon entropy of the probability distribution at the current step as $e = -\sum_{j=1}^{k} p(x_j) \log p(x_j)$;
- 3. The current proportion of tokens selected for watermarking in the generated sequence as $r = \frac{1}{n}\sum_{i=1}^{n} Selected(t_i)$

The probability of selecting the current token for watermarking is given by:

$$w_{\text{wm}} = \mathcal{M}_\theta\left([E,\ e,\ r]\right) \in [0,1],$$

During the detection phase, we determine whether each token in the generated sequence was likely selected for watermarking by re-evaluating the same signals used during generation. The sentence embedding $E$ can be recomputed based on the $k$ tokens before the current token, the Shannon entropy $e$ can be reconstructed using the model's token probability distribution at that position, and the watermark ratio $r$ can be tracked incrementally based on watermark decisions of the tokens before it. Since all three inputs to $\mathcal{M}_\theta$ can be obtained with prior sequence, the selection logic is fully reproducible during detection.

# B  Dataset details

We followed previous works [8, 9, 12, 13, 21] utilizing the RealNewsLike subset of the C4 dataset [29] as our dataset. Because the method we proposed requires training, we need to devide a training set as well as a test set for the experiments. We selected the first 10,000 pieces of text in the first RealNewsLike subset as training set. From another RealNewsLike subset, we randomly shuffled it using a random seed provided in our code. Subsequently, 500 samples of text with sufficient length were then filtered and selected to ensure that each is long enough be divided into a prompt segment and a reference text. They are later used as the test set in our experiments.

# C  Selective Watermarking Algorithm and Detection Algorithm of LTW

The following Algorithms show the detailed steps of how our selective watermark is applied and detected with the already trained Selector Network.

---

**Algorithm 1** Watermark Injection

---

1: **Input:** LLM $f$, prompt tokens $x$, window size $k$, green-list ratio $\gamma$, watermark bias $\delta$, max length $L$, Sentence Embedding model $\mathcal{E}$, our trained MLP $\mathcal{M}_\theta$, thresholds $\tau_{\text{low}}, \tau_{\text{mid}}, \tau_{\text{high}}$

2: **Output:**

3: $S \leftarrow x; \quad W \leftarrow []$                             *(sequence &generated sequence & watermark record)*

4: **for** $t = 1$ **to** $L$ **do**

5:      $l_t \leftarrow f(S)$                                         *(logits)*

6:      $p_t \leftarrow \text{softmax}(l_t)$

7:      $e_t \leftarrow -\sum_v p_t[v] \log p_t[v]$                          *(entropy)*

8:      $r_t \leftarrow \begin{cases} 0, & |W| = 0, \\ \text{mean}(W), & \text{otherwise} \end{cases}$

9:      $E_t \leftarrow \mathcal{E}(S[-k:])$            *(last k token for getting sentence embeddings)*

10:     $m_t \leftarrow \mathcal{M}_\theta([E_t, e_t, r_t])$                    *(selector output $\in [0,1]$)*

11:     $\tau_t \leftarrow \begin{cases} \tau_{\text{low}}, & r_t < \text{Low Ratio Threshold}, \\ \tau_{\text{high}}, & r_t > \text{High Ratio Threshold}, \\ \tau_{\text{mid}}, & \text{otherwise} \end{cases}$

12:     **if** $m_t > \tau_t$ **then**

13:        $m_t \leftarrow 1$

14:        $G_t \leftarrow \text{GreenMask}(l_t, \gamma)$

15:        **for** each $v \in G_t$ **do**

16:           $l_t[v] \leftarrow l_t[v] + \delta$

17:        **end for**

18:     **else**

19:        $m_t \leftarrow 0$

20:        $l_t[v] \leftarrow l_t[v]$

21:     **end if**

22:     Append $m_t$ to $W$

23:     $y_t \sim \text{Sample}(\text{softmax}(l_t))$

24:     Append $y_t$ to $S$

25:     **if** $y_t = \langle \text{EOS} \rangle$ **then break**

26: **end for**

27: **Return** $S$

---

---

**Algorithm 2** Selective Watermark Detection

---

1: **Input:**LLM $f$, prompt $x$, test sequence(prompt included) $y = \{y_0, \ldots, y_{N-1}\}$, window size $k$, green-list ratio $\gamma$, Sentence Embedding model $\mathcal{E}$, trained MLP $\mathcal{M}_\theta$, thresholds $\tau_{\text{low}}, \tau_{\text{mid}}, \tau_{\text{high}}$

2: **Output:** z-score $z$

3: $W \leftarrow []$;   $n_{\text{scored}} \leftarrow 0$;   $n_{\text{green}} \leftarrow 0$

4: **for** $t = |x|$ **to** $N - 1$ **do**

5:   $l_t \leftarrow f\big(y[0 : t]\big)$

6:   $p_t \leftarrow \text{softmax}(l_t)$

7:   $e_t \leftarrow -\sum_v p_t[v] \log p_t[v]$                                                              *(entropy)*

8:   $r_t \leftarrow \begin{cases} 0, & |W| = 0, \\ \text{mean}(W), & \text{otherwise} \end{cases}$

9:   $E_t \leftarrow \mathcal{E}\big(y[\max(0, t - k) : t]\big)$                                        *(sentence embedding)*

10:   $m_t \leftarrow \mathcal{M}_\theta\big([E_t, e_t, r_t]\big)$

11:   $\tau_t \leftarrow \begin{cases} \tau_{\text{low}}, & r_t < \text{LowRatio}, \\ \tau_{\text{high}}, & r_t > \text{HighRatio}, \\ \tau_{\text{mid}}, & \text{otherwise} \end{cases}$

12:   **if** $m_t > \tau_t$ **then**

13:     $m_t \leftarrow 1$;   $n_{\text{scored}} \leftarrow n_{\text{scored}} + 1$

14:     $G_t \leftarrow \text{GreenMask}\big(y_{t-1}, \gamma\big)$

15:     **if** $y[t] \in G_t$ **then**

16:       $n_{\text{green}} \leftarrow n_{\text{green}} + 1$

17:     **end if**

18:   **else**

19:     $m_t \leftarrow 0$

20:   **end if**

21:   Append $m_t$ to $W$

22: **end for**

23: $z \leftarrow \text{ZScore}\big(n_{\text{green}}, n_{\text{scored}}\big)$

24: **Return** $z$

---

# D   Further Discussions on Experiment Results

## D.1   ROC Curves

Figure 8 and 9 illustrate the ROC curves and the AUC values of our watermark methods and baseline methods under paraphrase attack. The result demonstrate the robustness of our methods compared to baseline methods.

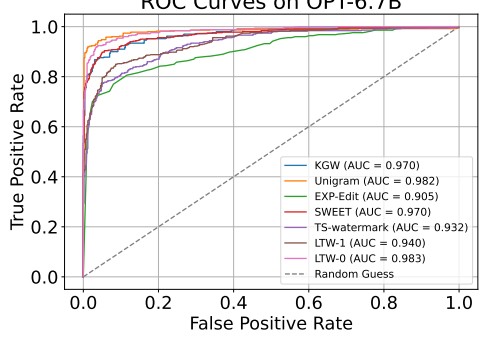

Figure 8:  Comparisons of ROC curves and AUC values of different watermark methods generated using OPT-6.7B when under dipper attack.

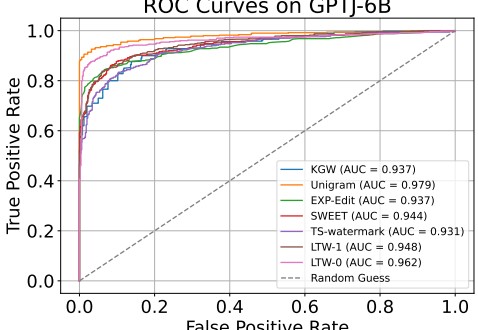

Figure 9:  Comparisons of ROC curves and AUC values of different watermark methods generated using GPT-J-6B when under dipper attack.

## D.2 Watermarked Text Example

We present an example of text watermarked using our method and baseline methods. Figure 10 shows the prompt used for generation using OPT-6.7B as well as the watermarked generations using our method, KGW and SWEET. The unwatermarked text is highlighted in gray, those highlighted in red or green are watermarked tokens, red represents the sampled token is in the red list, and green represents the sampled token is in the green list.

**Prompt**
Congrats to everyone featured in the videos and also to all of us, for being recognized as an Arts & Culture hub of the world! Of course we've all known this for a while now, but it's great that Google is catching on. If you're opening up this newsletter in your inbox for the first time or you're reading through stories on the web, you've stumbled across a new project called Maker City KC. This is our weekly newsletter. Thanks for finding us. Maker City KC is a place that highlights the city's many makers and entrepreneurs, creating an audience for the city's maker spaces and the spirit of innovation in our city. We depict and discuss the many dimensions of the 4th Industrial Revolution, and the opportunities it creates for Kansas City. And, as part of increasing civic engagement, we want ...

**KGW (z-score: 14.22, ppl: 20.85)**
If you don't have a newsletter, make sure to sign - up for one. What's going on in the world The International Date line is a place where the date that people on the surface see is different than the date that goes past the sun's per ist alt ic motion inside the earth, at some midnight, past the International Date Line, and back in to Kansas City, Missouri again. Well we had this crazy notion that if the International Date lines goes past Kansas City, our date for World Maker F aire would be in the past too â. World Maker F aire Kansas City will be happening this Friday - and our W F KC model is completely rooted and organized by YOU so please help spread the word about it! Create a T - shirt that is only Kansas City M akers! If you don't already have a shirt, make sure to get

**SWEET (z-score: 14.38, ppl: 17.64)**
Know of anyone that should be added to our list of makers, are a maker and would like to get some coverage in the weekly newsletter, let us know HERE. What You Should Know D rain our reservoir D rain Kansas City has launched in response to hundreds of thousands of illegal dump ers, who are cl ogging our reservoirs with the waste of others. As a city we should be fighting this instead of toler ating it. Visit drain our res erv oir. com to read up on the history of the problem the City has. Visit drain our res erv oir d ive in. com for a survey you can easily take and share it with your friends. Visit drain our res erv oir now. com for a map of what all the problems really are. Let the government know If you live in the City of Kansas City proper, please take the time to voice your concerns about what our government says on

**LTW (ours) (z-score: 14.67, ppl: 15.22)**
. This week's episode of The M akers Podcast features an episode of The M akers Podcast, produced by Dave De ans and Rebecca Miller at the Missouri History Museum. This week's guest is Brent Schmidt, CEO at City Fund KC. Our conversation with Brent spans City Fund's mission as a fund, through City Fund KC's ability to drive more entrepreneurship and economic development in Kansas City. Dave and Rebecca also talk about being in the Museum. Brent's background as a video producer and much more. This podcast is worth a listen. If you're in the market for some fresh, affordable, fresh, handmade, affordable gifts, make sure you're checking out Shop Now, an up start maker that sells 100 % handmade, fresh, affordable, gifts that you'll actually enjoy. This week's episode of The Maker Town Podcast features an

Figure 10: A real example of watermarked text generated on our C4 RealNewsLike test set. The figure shows comparison between of method and baseline method. In this example, the generated watermarked text using our method has a lower perplexity and a higher z-score than baseline methods.

## D.3 Correlation Between Entropy and Network Output

We conducted additional analyses by plotting a scatter diagram to investigate the relationship between the entropy of the input and the output of the trained model during selective watermarking. The results indicate a clear overall positive correlation between input and output entropy. Notably, when the entropy is either very low or very high, the results exhibit clustering within a narrow range. In these cases, other input factors, such as semantic embeddings, either amplify the influence of entropy or play a comparatively minor role in determining the outcome.

Specifically, when the entropy is below 1 or above 4, the output consistently falls into well-defined categories: low entropy values are classified as indicating that the current token should not be watermarked, while high entropy values are classified as requiring the application of the watermark. However, for entropy values within the range of 1.5 to 3.5, a broader spectrum of outputs is observed. Under our dynamic thresholding approach, the outputs under the same entropy includes the possible decision to selectively watermark or not applying the watermark to the token being generated.

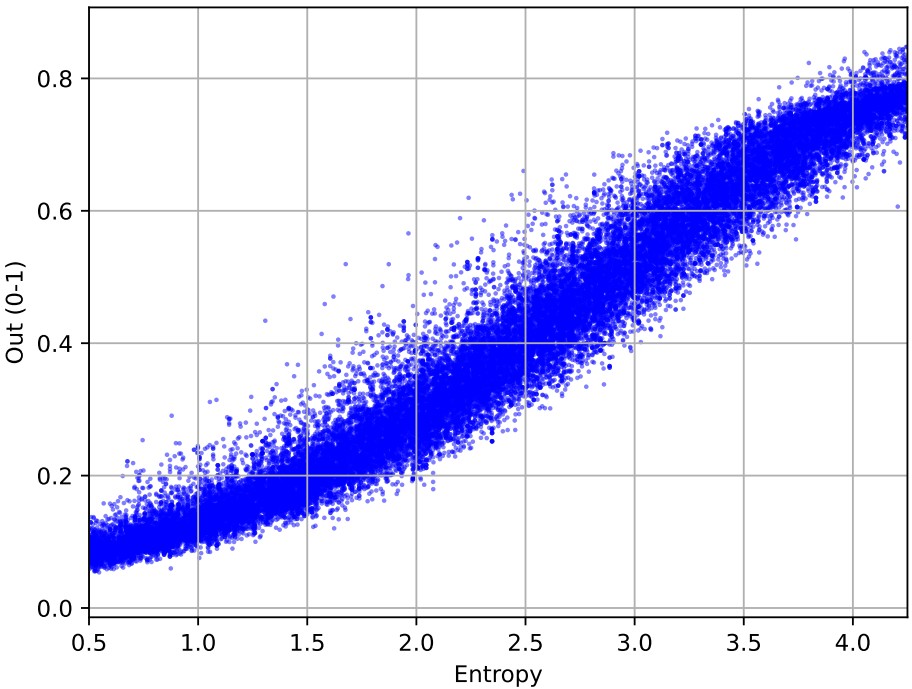

Figure 11: Scatter diagram between the entropy input and the output of our Selector Network.

## E    Further discussion on Selective watermark designs

To compute the final selection weight $m_{\mathrm{wm}} \in [0, 1]$, we design a compact yet expressive neural network based on a multi-layer perceptron (MLP) [33]. The input to the model consists of a sentence embedding of dimension 768, along with two input features: entropy and ratio. In order to handle the high-dimensional nature of the embedding and promote efficient learning, the embedding is first passed through several fully connected layers to reduce its dimensionality. This reduced representation is then concatenated with the two features to form a concatenated input vector, which is subsequently fed into another MLP responsible for computing the final prediction. The output layer applies a sigmoid activation function to constrain the result to the range $[0, 1]$. The model functions as a selector module in our system, leveraging both semantic information from the sentence embedding and statistical cues from the two watermarking related features to make a learnable, data-driven selection decision.

## F    Experimental Details

### F.1    Training Details

In our proposed Selector Network, implemented as a multi-layer perceptron (MLP), we adopt a two-stage processing pipeline for the input. Specifically, each input sentence embedding $E \in \mathbb{R}^{768}$ is

first passed through a two-layer MLP for dimensionality reduction. The resulting lower-dimensional representation is then concatenated with the other two additional features: the entropy $e \in \mathbb{R}$ of the predicted token distribution and the watermarked ratio $r \in [0, 1]$. Letting $E_r$ denote the reduced embedding, the concatenated vector $[E_r; e; r]$ is used as input to a second two-layer MLP, which produces a single output in the output layer. All hidden layers in both MLP modules use the Leaky ReLU activation function. To ensure the output lies in the interval $[0, 1]$, the output layer uses a sigmoid activation function. We train the Selector Network for one epoch on our training set. We used OPT-1.3B during training, generating a maximum of 75 tokens for each prompt provided during training. During training, the parameters in the LLM as well as the sentence embedding model are frozen, the parameters in our trained MLP are the only parameters being updated. The training procedure uses a batch size of 5, the Adam optimizer with a learning rate of $1 \times 10^{-4}$, and input sentence embeddings computed from the previous $k = 6$ tokens in the previous sequence. The watermark hyperparameters are set to $\delta = 3.0$ and $\gamma = 0.25$, the same as those in the main experiments. We save model checkpoints every 200 steps and retain the final model weights after one epoch for use in experiments on the effectiveness of our method.

## F.2 Experiment Details

In our main experiments, we set $\gamma = 0.25$ and $\delta = 3.0$ for methods that can adjust these hyperparameters, and we set the entropy threshold as 1.2 for SWEET. We used the default model provided in their code for TS-watermark, and for EXP-edit, the default hyperparameters provided in their code. For all watermarks, we used Multinomial Sampling with a top-k of 100 a top-p of 0.95 and a no repeat n-gram size of 8 to generate 200±25 tokens for each prompt.

For fair and consistent comparison, we uniformly set the tunable hyperparameters $\delta = 3.0$ and $\gamma = 0.25$ across all watermarking schemes that allow manual configuration of these values in our main experiment. This decision was motivated by the hyperparameter configuration adopted in the baseline method SWEET, which introduces an additional hyperparameter, entropy, to determine whether to add watermark at each step. The authors conducted a comprehensive hyperparameter search and ultimately selected $entropy = 1.2$ with $\delta = 3.0$ and $\gamma = 0.25$ as their main setting which was the only set of hyperparameter that achieved both high detectability and high text quality under their predefined criteria on one of the evaluated datasets. The setting $\delta = 3.0$ and $\gamma = 0.25$ was adopted as the primary configuration throughout their main and subsequent experiments in their work.

Consequently, we fix entropy to 1.2 for the SWEET baseline and apply $\delta = 3.0$ and $\gamma = 0.25$ to all applicable watermarking methods in our evaluation. In the context of our main experiment, we consider a watermarking approach to be inferior to ours under the current evaluation setup if: (1) it does not outperform ours in either detectability or robustness (considering some metrics where all methods perform comparably, and (2) it yields lower text quality.

One of the fundamental advantages of large language model (LLM) watermarking lies in its high detectability under sufficient watermark strength. Our proposed watermarking method exhibits excellent detectability: on a 500-sample test set all watermarked samples were successfully identified, with no false positives on unmarked texts. We also did extra experiments to compare our method with KGW under various watermark strengths varying from $\delta = 1.0$ to $\delta = 3.5$ to evaluate the performance of our watermark with the baseline method in terms of detectability and text quality, the results show that our methods have better pareto frontier than theirs.

The improved robustness of LTW-0 can be attributed to its fixed green-red token partitioning strategy, which mitigates failure cases in using a hash-function that is based on previous generated text, where rewriting of previous tokens affects green/red list assignment and causes detection to fail. The implementation of LTW-0 in our work is simple, the difference between our implementation of LTW-1 and it is a changed function for obtaining the green-list, changing the hashing method to using a fix hash-key. We adopt the same hash-key that was used in previous works. [12, 17].

## F.3 Computational Resources and Runtimes

We performed our experiments on Nvidia L20 GPUs. We trained the our network using 3 L20 GPUs, among which the second and third GPU are mainly used to speed up the generation of the LLM. It takes about 60 hours for training 1 epoch. In our other experiments, evaluating the performance of

our watermark method and baseline methods, only one L20 is used, without using parallelism. About 2 hours is needed for finishing the watermarking generation task for our method, baseline methods expect EXP-edit, which is slow requires about the same time for finishing generation watermarked text using the test set.

## G   Discussion on Multiple-Gradient Descent Algorithm

To train the network we proposed, we formulated two loss functions $\mathcal{L}_Q$ and $\mathcal{L}_D$ to describe our task. In our work, the training objective is to minimize those two loss functions, $L_Q(G)$ and $L_D(G)$. However these two losses often exhibit a competitive relationship, meaning that reducing one may lead to an increase in the other. The optimization problem is formulated as:

$$\min_G L_Q(G) \quad \text{and} \quad \min_G L_D(G)$$

This is a typical multi-objective optimization problem, where the ideal solutions are characterized by Pareto optimality.

Given two feasible solutions $G$ and $\overline{G}$, we say that $G$ dominates $\overline{G}$ if:

$$L_Q(G) \leq L_Q(\overline{G}), \quad L_D(G) \leq L_D(\overline{G}),$$

and at least one of these inequalities is strict. A solution $G^*$ is called Pareto optimal if there does not exist another solution that dominates $G^*$.

The Multiple Gradient Descent Algorithm (MGDA) [5, 35] is designed to find a descent direction that leads the optimization towards the Pareto front. Specifically, we first compute the gradients of $L_Q$ and $L_D$ with respect to $G$, denoted as $g_Q$ and $g_D$, respectively. MGDA seeks a direction $g$ that is a convex combination of $g_Q$ and $g_D$:

$$g = \lambda^* g_D + (1 - \lambda^*) g_Q,$$

where $\lambda^* \in [0, 1]$ is chosen to minimize the norm of $g$:

$$\lambda^* = \text{argmin}_{\lambda \in [0,1]} \|\lambda g_D + (1 - \lambda) g_Q\|_2.$$

Previous works [35] suggests, an approach is applicable in the case of two losses, where the Frank-Wolfe algorithm  [10] can be effectively utilized since the line search step admits a closed-form solution.

---
**Algorithm 3** Closed-form solution for $\lambda^*$

---
**Require:** Gradients $g_Q$, $g_D$
1: **if** $g_D^\top g_Q \geq g_D^\top g_D$ **then**
2:     $\lambda^* = 1$
3: **else if** $g_D^\top g_Q \geq g_Q^\top g_Q$ **then**
4:     $\lambda^* = 0$
5: **else**
6:     $\lambda^* = \dfrac{(g_Q - g_D)^\top g_Q}{\|g_D - g_Q\|^2}$
7: **end if**

---

By updating the model parameters along the direction $g$, the optimization process is guided towards solutions that are Pareto optimal with respect to both $L_Q$ and $L_D$.

## H   Discussion about Limitations and Future Work

Our method presents two main limitations, primarily due to constraints in computational resources and time.

First, due to limitations in computational resources and time, our experiments are conducted on relatively small-scale models and only on one dataset. Specifically, we utilize the RealNewsLike

subset of the C4 dataset which has been widely adopted in previous works [8, 9, 12, 13, 21] on large language model watermarking.

Second, the selector network in our work is trained on generations produced by the OPT-1.3B model using C4 RealNewsLike subset as input for evaluating our method with baseline on the text continuation task. In future work, we may extend this framework to broader scenarios which may require adapting the selector to different generation tasks, which may need to train our network on different tasks using LLMs which may experts in those tasks.

