# OpenReview forum: "Learning to Watermark: A Selective Watermarking Framework for Large Language Models via Multi-Objective Optimization"
_NeurIPS.cc/2025/Conference — NeurIPS 2025 poster_

### Official Review · Reviewer_9Hqn · 2025-07-01

**Clarity:** 3
**Significance:** 2
**Originality:** 2
**Rating:** 4
**Confidence:** 3

**Summary:**

This paper trains a neural network to achieve the selective watermarking strategy by analyzing sentence embeddings, token entropy, and the current watermarking ratio, which achieves a better balance between text quality and detectability.

**Questions:**

Please see above.

**Ethical Concerns:**

["NO or VERY MINOR ethics concerns only"]

**Final Justification:**

In general, most of the concerns have been resolved.

**Limitations:**

Please see above.

**Paper Formatting Concerns:**

No Paper Formatting Concerns.

**Quality:**

2

**Strengths And Weaknesses:**

Strengths:

1. This paper provides an optimized method to achieve a selective watermarking approach.

2. This paper provides extensive experiments to demonstrate the effectiveness of the proposed methods.

Weaknesses:

1. Could the authors consider using additional datasets, such as long-form QA, to further validate the effectiveness of the proposed method?

2. OPT-6.7b is too old compared with the most recent open-source LLM, like Llama-3.

3. Table 1 and 2 provide a performance comparison of different watermark methods. Why does Table 2 not provide TPR@2? Moreover, this paper strengthens the trade-off between the text quality and detectability. One better way to demonstrate this trade-off is to provide TPR with a lower FPR, such as 1%, 0.1%, 0.01%.

4. Could the authors explain the reason for using the similarity metric? It seems that this metric can not accurately represent the quality of watermarked text, because it is not necessary to have a high similarity between watermarked text and reference text especially for the open generation dataset (C4).

5. Is it possible to report the rate of watermarked tokens?

---

> ### Author Rebuttal · Authors · 2025-07-29
>
> **Reply to Reviewer 9Hqn**
>
> Thank you for your valuable and insightful comments. We will explain your concerns and questions point by point. In addition, we will provide you with **the results of the additional experiment which may address your concern**. We hope that our rebuttal could properly address your concerns and questions.  If so, we would deeply appreciate it if you could kindly raise your score (Rating: 3: Borderline reject), If not, please let us know your further concerns. Thank you for your invaluable time and consideration.
>
> > **Q1:** Could the authors consider using additional datasets, such as long-form QA, to further validate the effectiveness of the proposed method?
>
> **Reply:** We thank you for your advice of suggesting us to use additional  long-form QA dataset to further validate the effectiveness of the proposed method, which will be provided in the following additional experiment part using the ELI5 dataset. The original dataset, C4 realnewslike dataset was frequently used in previous works [1-5] of LLM watermarking ,sometimes the only dataset used in the paper ,thus we used it for our original experiments.
>
> [1] A Watermark for Large Language Models，ICML 2023
>
> [2] Context-aware Watermark with Semantic Balanced Green-red Lists for Large Language Models. emnlp 2024
>
> [3]  Robust distortion-free watermarks for language models,TMLR
>
> [4]  Token-Specific Watermarking with Enhanced Detectability and Semantic Coherence for Large Language Models,ICML 2024
>
> [5] Adaptive Text Watermark for Large Language Models ,ICML2024
>
> ---
>
>
>
> > **Q2:** OPT-6.7b is too old compared with the most recent open-source LLM, like Llama-3.
>
> We thank you for your advice of suggesting that we use more recent open-source LLM such like Llama3,  we will provide the results of using it in our additional experiments.  We have to point out that we have chosen the OPT model is because it is one of the most frequently used model in the works[1-5] on watermarking ever since the KGW watermark[1] , and thus is a reason we have included it.
>
> [1] A Watermark for Large Language Models，ICML 2023
>
> [2] Context-aware Watermark with Semantic Balanced Green-red Lists for Large Language Models. emnlp 2024
>
> [3] Robust distortion-free watermarks for language models,TMLR
>
> [4] Token-Specific Watermarking with Enhanced Detectability and Semantic Coherence for Large Language Models,ICML 2024
>
> [5] Adaptive Text Watermark for Large Language Models ,ICML2024
>
>
>
> **Additional experiments to provide a more comprehensive evaluation of LTW:**
>
> We appreciate the reviewer's insightful observation regarding the need for comparisons using additional dataset and additional model to evaluate the effectiveness of our watermark method. **We genuinely value the opportunity to clarify and address your concerns. Taking advantage of the rebuttal period, we have conducted additional experiments to provide a more comprehensive evaluation of our watermarking method and further validate the effectiveness of the proposed method.** We have adopted the ELI5 dataset which is a long form Question Answering dataset, we randomly selected 500 piece of question-answer pairs from the dataset.
>
> We employed the selective watermarking network previously trained on the C4 dataset. Due to the limited time available for the rebuttal period, we were unable to retrain a network using data from the QA dataset which would undoubtedly  bring better performance. However, the model trained on C4 demonstrates sufficient generalization ability, using it is sufficient for the limited time being. In evaluating detectability, we followed Reviewer GajM’s suggestion and adopted the use of p-values under a small false alarm rate, which provides a more rigorous measure of detectability instead of more commonly used metrics in watermarking papers such as FPR@1 and FPR@2 . As for text quality, we used the recall score of of ROUGE-L, a simpler and more computationally efficient metric.
>
> | Method       | LLaMA3.1-8B    |           | OPT-6.7B        |           |
> | ------------ | -------------- | :-------- | --------------- | --------- |
> |              | p-value < 1e-8 | Rouge-L   | p-value < 1e-10 | Rouge-L   |
> | KGW          | 0.984          | 0.230     | 0.994           | 0.244     |
> | Sweet        | 0.986          | 0.221     | **0.996**       | 0.252     |
> | TS-watermark | 0.934          | 0.225     | 0.992           | 0.251     |
> | LTW-1 (ours) | 0.982          | **0.251** | 0.992           | **0.256** |
> | LTW-0 (ours) | **0.988**      | 0.242     | 0.994           | 0.249     |
>
>  Our experimental results in the table demonstrate that our method demonstrate high detectability as well as text quality. Specifically, while Sweet has a performance slightly better than ours in terms of detectability when generated using OPT-6.7B on this task, our methods shows better overall performance, **especially the text quality when generated using LLaMA3.1-8B under this task**. Compared with TS-watermark ,which was trained to adaptively provide $\\gamma$ and $\\delta$ hyperparameters using the C4 dataset, our method shows better generalization ability .
>
> ---
>
>
>
> > **Q3:**  Why does Table 2 not provide TPR@2? Moreover, this paper strengthens the trade-off between the text quality and detectability. One better way to demonstrate this trade-off is to provide TPR with a lower FPR, such as 1%, 0.1%, 0.01%.
>
> **Reply:** We used tpr@2 because metrics like tpr@1 and tpr@2 are frequently used in previous works ,for Table 1,the results of using tpr@1 are as follows shows, only the metrics of EXP-edit and TS-watermark became lower. As for Table 2, we didn't provide tpr@2 because at 2% fpr they all achieved 100% tpr when not under attack, this can also be observed from the AUROC metrics value. In the additional experiment, however, we will set a low probability of false-alarm,such as 1e-8, and use $p-value < P_{FA}$ as the metric in the additional experiments to show the high detectability our method possess.
>
> |       | kgw-1 | unigram | EXP-edit | sweet | TS-watermark | LTW-1 | LTW-0 |
> | ----- | ----- | ------- | -------- | ----- | ------------ | ----- | ----- |
> | tpr@1 | 0.998 | 1.000   | 0.854    | 1.000 | 0.992        | 1.000 | 1.000 |
>
> ---
>
>
>
> > **Q4:** Could the authors explain the reason for using the similarity metric?
>
> **Reply:** We understand the reviewer's  question over the similarity metric. In this paper, we used  the cosine Similarity between the sentence embeddings for assessing the text quality and semantic coherence between the generated text and the original part that was used as reference text. In the work of one of our baseline methods, TS-Watermark[4] ,also adopted similar approach for evaluate the semantic coherence of the watermarked text under the C4 dataset.  In our case, we use data from the C4 realnewslike subset, which are more like news articles for the LLMs to generating continuation writing based on the part of text that is given. A higher similarity indicates a more contextually appropriate and more coherent connection with the preceding context.
>
> ---
>
>
>
> > **Q5:** Is it possible to report the rate of watermarked tokens?
>
> **Reply:** We appreciate the reviewer's concerns regarding the rate of the  watermarked tokens, the average watermarked percentage  of text generated under the c4 realnewslike subset is about 36.73%, while those generated under the ELI5 dataset is about 47.35%.  This may be due to the difference between the tasks.

---

> > ### Comment · Reviewer_9Hqn · 2025-08-05
> >
> > I appreciate the authors for the additional experiments and discussion. Specifically, for Q3, authors could refer to the experiments of [1], which generate 100k texts to evaluate the TPR@1e-6FPR, instead of using the method mentioned in the Q3 response. Of course, I understand the time limitation to conduct such an experiment, but it is encouraged to have this kind of experiment in the finalized version to better demonstrate the authors' claim. This is because the difference among different methods is not obvious when the FPR is high.
> >
> > In general, most of the concerns have been resolved, and I will increase my score correspondingly.
> >
> > [1] LLM watermark detectability-robustness-quality tradeoff. In The Thirty-eighth Annual Conference on Neural
> > Information Processing Systems, 2024.

---

> > > ### Author Response · Authors · 2025-08-06
> > >
> > > Thank you so much for your kind feedback. We truly appreciate your positive and insightful response as well as the improved rating. Your encouragement motivates us to do even better—thank you again！

---

### Official Review · Reviewer_v5TR · 2025-07-02

**Clarity:** 2
**Significance:** 2
**Originality:** 3
**Rating:** 4
**Confidence:** 5

**Summary:**

This paper introduces Learning to Watermark (LTW), a novel selective watermarking framework for large language models (LLMs). The method uses a lightweight neural network (Selector Network) trained via multi-objective optimization to decide dynamically when to watermark tokens during generation. By incorporating sentence embeddings, token entropy, and the watermark ratio, LTW aims to balance text quality and detectability more effectively than prior approaches.

**Questions:**

Can the selector network trained on one LLM generalize to other LLMs? In line 278, the text refers to Table 3 and Table 4, but Table 4 is missing from the paper. In Tables 1 and 2, the number of decimal places is inconsistent. Could the authors explain the reason for this formatting choice?

**Ethical Concerns:**

["NO or VERY MINOR ethics concerns only"]

**Limitations:**

1. The method emphasizes the need for an adaptive threshold to decide whether to watermark a given token. However, Table 3 shows that the gains in detectability and quality from this adaptive threshold are quite limited. This raises questions about whether this component is really necessary.

2. Since LTW requires an additional sentence embedding model and selector network during both generation and detection, it introduces higher computational and memory overhead. The authors should provide a discussion and experiments on the time and space costs of the approach.

3. From the proposed method, it appears that the watermark ratio is adjusted based on the portion of text already generated. This results in a higher watermark ratio in the earlier parts of the text and a lower ratio in later parts. This design may naturally expose the watermark to truncation attacks, where keeping only the later segments could reduce detectability. Could the authors clarify whether this would significantly impact detection performance?

**Quality:**

3

**Strengths And Weaknesses:**

**Strengths:**
1. The paper is clearly written and well-structured and easy to follow.
2. LTW proposes learning selective watermarking to balance text quality and detectability, going beyond entropy thresholds and introducing adaptive criteria based on richer contextual signals.

**Weaknesses:**
1. Eq.5 shows that the objective is to maximize the semantic similarity between the watermarked and unwatermarked text. However, it is unclear what exactly the unwatermarked text refers to and how it is generated in practice. The authors should clarify this point to avoid confusion about the optimization target.
2. The paper mentions that LTW achieves strong robustness. Why is this not discussed in the introduction? More importantly, why does LTW improve robustness?  For example, Table 1 shows that LTW achieves higher robustness than KGW, but the reasons for this advantage are not clearly discussed.
3. According to Table 1, Table 2, Figure 8, and Figure 9, LTW outperforms KGW in both detectability and text quality. Yet from Equations 3 and 7, LTW seems like a relaxed version of KGW. Why does this weaker formulation yield a better results?
4. There is limited discussion and no ablation study on the impact of the losses defined in Eq.(10) and Eq.(11), which are central to the optimization process.

---

> ### Author Rebuttal · Authors · 2025-07-29
>
> **Reply to Reviewer v5TR**
>
> Thank you for your thorough review and for pointing out the Table number reference issues. It was a reference mistake that we will correct. We are deeply grateful for the time and effort you have invested in reviewing our paper. To address your queries and concerns, we have prepared detailed responses for your questions and concerns and **we genuinely value the attention you give to our work.**
>
> > **Q1:** The authors should clarify what exactly the unwatermarked text refers to and how it is generated in practice to avoid confusion about the optimization target.
>
> **Reply:** We apologize for any misunderstanding that may have arisen regarding the meaning of the unwatermarked text and we will further explain it and how it is generated. In Eq.5 we aim to design a metric to quantify the impact of watermarking on generation quality compared to non-watermarked outputs generated by the LLM. By optimizing this objective during training, we seek to minimize the degradation introduced by watermarking. Naturally, one effective approach is to compute the cosine similarity between the generated embedding of the tokens that were generated without the watermark and that of the watermarked one. Therefore, during training, we generate both the non-watermarked and watermarked embeddings for the same prompt to enable this comparison. The cosine similarity is then computed and used as a loss term, encouraging the watermarked content to remain semantically closer to its non-watermarked counterpart.
>
> ---
>
>
>
> > **Q2:** Why is robustness not discussed in the introduction? More importantly, why does LTW improve robustness?
>
> **Reply:** We appreciate your concerns and questions about robustness which we didn't explain clearly in the paper due to page limitations. We genuinely value the opportunity to clarify and address your concerns about robustness. We will explain **the measures during watermark design to prevent ill Impact on robustness step by step** :
>
> Reviewer GajM asked us out of *Curiosity* that why have we chosen a hard binary mask instead of keeping the soft weights of the selector network like during training, which can be more flexible. This is a design for robustness. For the soft mask used during training,  the mask is also a detection weight as in Eq.7, a token with a small mask is also give light detection weight, however, though this brings more flexible may be potentially better in terms of text quality. We believe that, comparatively, using the hard mask can better enhance the robustness of the watermarking scheme under paraphrasing attacks. From a design perspective, it is relatively simpler to categorize the outputs of the selector into two distinct classes—"apply watermark" and "do not apply watermark"—and apply watermarking accordingly. During detection, we only need to verify the outputs that were marked for watermarking, which is a simple approach.
>
> However, if a soft mask, as used during training, were to be adopted at inference time, the detector would likewise need to consider the soft mask values as per-token detection weights. In such a case, tokens with very low mask values would contribute minimally to detection, while the opposite for those with a high mask value. While this offers finer granularity, it introduces more complexity and uncertainty—especially more vulnerable under paraphrasing attacks.
>
> The inputs of our selector includes a suitable window size k for obtaining the sentence embedding, using a k that is too short or if simply using the previous token embedding which would be easier to obtain and no need for the extra sentence would make it vulnerable  to paraphrasing attacks since the previous may change due to the attack. Previous work[1] have shown that sentence semantic meanings are robust under paraphrasing attacks.(In their work ,they replaced KGW's[2] hashing method with a semantic one, improving robustness). This is a method done to enhance the robustness of our network output.
>
> Paraphrasing attacks is hard to change "apply watermark" to "do not apply watermark" than changing its values. In our Eq.9, we penalizing those un-masklike outputs, making most outputs further form the threshold. In contrast, soft mask values are **continuous and more sensitive** to subtle changes, meaning that a paraphrasing attack may cause small shifts in mask values (e.g., a slightly increased value for a previously low-scored token, or a slightly decreased one for a previously high-scored token). These small perturbations can **accumulate** and increase the **uncertainty of watermark detection**, thus reducing robustness.
>
> These are some of the measures taken during watermark design to prevent ill impact on robustness, we hope this helps you better understand the measures we have taken in our design to enhance robustness though they may not be something novel enough to boast about.
>
>  [1] A Semantic Invariant Robust Watermark for Large Language Models，ICLR2024
>
>  [2] A Watermark for Large Language Models，ICML 2023
>
> ---
>
>
>
> > **Q3:**  Why does this weaker formulation yield a better results?
>
> **Reply:** We need to point out that a relaxed formula doesn't mean that it would provide less results, our relaxed version is designed to selectively apply and detect watermark based on a "rule", which is the output of the selector network. The previous selective watermark, Sweet[1] , can also be seen as a relaxed version of KGW. The rule of which is whether the token has passed the entropy threshold. The method has achieved better coding ability and detectability than KGW though it is also a relaxed formulation.
>
> We believe the reason that our methods are that they can grasp the tokens for which watermarking can bring more benefit. For example, taking entropy as an example, when the entropy is low, the model’s output distribution becomes highly concentrated on a few specific tokens. In such cases, watermarking is more prone to introducing errors, as perturbing the logits may lead to the selection of low-probability tokens that were unlikely to be chosen otherwise. Additionally, due to the sharp concentration of probabilities, bias added  to tokens in the green list tends to be more insignificant, as these tokens already have low baseline probabilities compared to the original high-probability ones . As a result, the sampling process remains dominated by high-probability tokens—which are often located in the red list (as the choice of the gamma value typically is smaller than 0.5). This increases the likelihood that the sampled tokens are in the red-list, thereby reducing detectability. We believe that our method can take grasping when to  make the decision watermark further, watermarking those that improves detectability and afflicts text quality less.
>
>  [1] Who Wrote this Code? Watermarking for Code Generation, ACL 2024
>
> ---
>
>
>
> > **Q4:** Can the selector network trained on one LLM generalize to other LLMs?
>
> **Reply:**  Yes ,The main results of our method using GPT-J used the selector trained using the OPT model. Part of the reason we used a separate sentence embedding model and obtain its sentence embedding is so that the network trained on one model can generalize to other LLMs. The results show that it can generalize to other LLMs.
>
> ---
>
>
>
> > **Q5:** The authors should provide a discussion and experiments on the time and space costs of the approach.
>
> **Reply:** The additional memory introduced by our watermarking method primarily stems from the use of an extra sentence embedding model and a lightweight selective neural network. In practice, the watermark generation process incurs approximately **0.5 GB** of additional GPU memory consumption, most of which is attributed to the sentence embedding model.
>
> As for inference time, we report  generated 100 samples using various watermarking methods for exactly 200 tokens, here is average time required for each piece of text:
>
> - **Ours (LTW)**: 6.25 seconds / text
> - **KGW**: 5.81 seconds / text
> - **SWEET**: 5.93 seconds / text
> - **TS-watermark**: 6.04 seconds / text
> - **EXP-edit**: 34.80 seconds / text
>
> ---
>
>
>
> > **Q6:**  This design may naturally expose the watermark to truncation attacks, where keeping only the later segments could reduce detectability. Could the authors clarify whether this would significantly impact detection performance?
>
> **Reply**: Your suggestion is very professional, and we appreciate your insight In addition to our theoretical analysis, we have conducted the following supplementary experiment to address it.
>
> In our watermarking method, we introduce the watermarking ratio as a controllable factor not only for the reasons stated in the paper, but also because we aim to design a watermarking approach that is more practical and generalizable than SWEET for real-world applications. Specifically, unlike SWEET, which requires task-specific hyperparameter tuning (e.g., determining the optimal entropy threshold through extensive search) prior to deployment, our method is designed to work without such tuning, making it more plug-and-play.
>
> The watermarking ratio serves as a balancing mechanism between entropy and semantics. It allows us to encourage watermarking when the overall ratio is too low and discourage it when the ratio is too high—rather than merely encouraging watermarking indiscriminately.
>
> To verify this, we recorded 500 generations using our watermarking method and tracked whether each generated token was watermarked. For each generation, we generated 200 tokens and split them into the **first 100 tokens** and the **last 100 tokens**. We then calculated and compared the proportion of tokens that were watermarked in both segments. The results show that the first half had a watermarking ratio of **38%**, while the second half had **35.4%**, indicating that no significant imbalance occurred for the first half and the later part. Thus, doing a simple truncation may not bring too much impact.

---

> > ### Comment · Reviewer_v5TR · 2025-08-06
> >
> > Thank you very much for the response. It has addressed my concerns well. Based on the authors’ reply, it appears that many of the performance gains of LTW rely on carefully balancing the ratios between different loss terms (as discussed in Q2, Q3, and Q6). If the authors could provide ablation studies on the components in Eq. (10) and Eq. (11), it would help better highlight the contributions of the paper. I understand that due to the time constraints of the rebuttal process, it may not be feasible to include additional experiments at this stage. I encourage the authors to consider adding more discussions and ablation studies in the revised version.
> >
> > Overall, I maintain my current score.

---

### Official Review · Reviewer_GajM · 2025-07-02

**Clarity:** 3
**Significance:** 3
**Originality:** 2
**Rating:** 4
**Confidence:** 4

**Summary:**

In this paper, the authors propose a methodology to improve Green-list based watermarking schemes for LLMs. The method is based on the design of a so-called "selector network", that is, a regressor mapping a series of features to a mask. The main idea is to prevent watermarking during steps where the token distribution is almost deterministic and thus would lead to both a high quality loss and few gains in detectability if any.

To achieve this, the authors leverage a multi-objective optimization objective, balancing quality and detectability and claiming to achieve Pareto optimality. Quality is taken into account using embedding similarity (between watermarked and non-watermarked sentence), entropy of

Finally, the authors compare their scheme to other classical methods, as well as comparable methods based on the entropy.

**Questions:**

1. Could the authors improve their evaluation by focusing on the TPR at a low FPR (at least less $10^{-5}$ ) instead of providing many insignificant metrics such as AUC, F1 etc..? The best course of action would be to switch to non-asymptotic p-values if times and resources allow it -- since I understand this would entail changing the loss of the selector network.
2. I would forego EXP-Edit in favor of another distortion-free scheme which can actually provide theoretical guarantees
3. Can you provide a rationale for the choice of LLMs and maybe provide an empirical analysis of the distribution of their entropy for each generation step and for different texts following my "**Uninformed choice of LLM**" comment ?
4. An empirical analysis of the probability of drawing a green token under $\mathcal{H}_1$ with and without the selector token would be welcome to get a better insight on the impact of the selector network on detection.
5. *(Curiosity)* Why have you chosen a hard binary mask instead of keeping the soft weights of the selector network? I would guess a soft mask to be more flexible, allowing to tune the bias $\delta$ automatically for a given distribution. Have you tried it? If yes, was it worse than the binary mask?

**Score increase or decrease** : Given the current state of the paper, I see no reason to decrease the score since I am confident in the author's results and since the methodology looks sound to me. I will definitely increase the score if: 1) better metrics are provided (most important), 2) a better selection of LLM is made with regard to their entropy (if time and resources permit) and 3) the proposed improvements to the experimental settings are included.

[3] S. Aaronson and H. Kirchner. Watermarking GPT outputs, 2023. URL https://
scottaaronson.blog/?m=202302.

**Ethical Concerns:**

["NO or VERY MINOR ethics concerns only"]

**Final Justification:**

I increase my score due to the fact that the authors have diligently answered all my questions. Citing my own comments:

> From what I can see, the TPR @ small FPR is still very high for every method. It is once again difficult for me to see any difference between either of the proposed scheme. The only two reasons I can think of is either too high of a temperature of too long of a text (how long is the text by the way?). This is why I would have appreciated to have an average p-value instead of a TPR, this would have allowed a better discrimination between the different schemes. Similarly, the ROUGE score differences do not seem significant to me.

> For the analysis of $\gamma$, I was just asking the author to compute the actual/empirical $\gamma^{\prime}$ on a sample of texts. This would have actually been a great way to compare schemes (since a higher $\gamma^{\prime}$ leads to lower p-values by definition.

> For Q4, I understand the choice from the authors. I believe they could have kept a detector which takes a 0-1 decision (does the token belongs to the green-list) while keeping the soft mask during the embedding to improve text quality. But maybe I am mistaken.

Consequently, the experimental design, though now better, is still somewhat lacking. As as such, I cannot go as high as an accept, but will gladly raise my score to a Borderline accept.

**Limitations:**

Nothing to report

**Paper Formatting Concerns:**

Reference error -> [13] should be https://arxiv.org/abs/2307.15593 not KGW

**Quality:**

2

**Strengths And Weaknesses:**

## Qualities

**Clear and well-written paper**: Each step of the method is well motivated and articulated. Each loss is well described. I was able to answer most my questions during second and third read of the paper. I am confident I should be able to try replicating the results with the information contained in the paper (and in the appendix for the MLP training details).

**Pertinent choice for the losses**: The collection of features and loss design for the selector network looks very complete and well-designed to me, taking into account all necessary properties for a watermark that is both robust and imperceptible.

**Nice analysis of the selector outputs**: I find Figure 7 and its corresponding paragraph very insightful. I appreciate the time taken to study the function the words chosen by the selector network.

## Weakness

**Detection score of Green-list tokens is flawed**: The authors use the original z-score of the KGW method as their basis for detection. This assumes asymptotic normality of the score and a text long enough that normality can be invoked to compute sound statistics (p-value and confidence interval). Such assumptions are known to be unnecessary, as shown in [1]. Indeed, the sum of token scores for green-list watermarks follow a binomial distribution $\mathcal{B}(\gamma, l)$ under $\mathcal{H}_0$ ($\gamma$ being the probability of the token to be in the green list for non-watermarked text and $l$ the size of the text). This means an **exact, non-asymptotic** p-value can be computed for KGW and Unigram, leading to a precise control over Type I errors. Currently the authors have to rely on empirical measurements which are less rigorous.

**Flawed choice of metrics**: This is a direct consequence of the previous problem. The authors use F1, AUC and TPR at (very!) high FPR. Obviously, watermarking schemes are good enough so that the detectability under no attack is always close to 1 for each metric. This makes comparison difficult and variations between schemes insignificant in Table 1. A better way to compare the schemes would be to fix a **low** probability of false-alarm, say $10^{-6}$, and compute the power of the test at this guaranteed rate. This can be done easily if a theoretically sound score is used during detection. In particular, using the $p$-value approach I outlined in the previous section allows to do this analysis without any problem. See any of [1,2] for a more thorough description of the method. Also note that $10^{-6}$ might still be too high a $P_{FA}$ to get significant difference in powers between watermarking schemes, in which case, I would advise the authors to go even lower.

**Uninformed choice of LLM**: Two LLMs are studied, OPT-6.7B and GPT-J-6B. No rationale is provided for why these models have been chosen. It would have been a lot more pertinent to choose three different LLMs, one with low, medium and high entropy on average and see the difference in gains using the authors method. I would make the author's method even more enticing as I expect the performance to be high for low entropy models. For an idea of what kind of LLM to choose, I know [1] has performed such a study in Appendix J: *Phi* models have notoriously high entropy, *Llama-2* very low entropy and *Llama-3.1* is between the two. At the very least, an analysis of the distribution of the entropy of currently used models should be provided

**Lack of link to theoretical analysis**: The distribution of the score for KGW follows a binomial distribution under $\mathcal{B}(\gamma^{\prime}, l)$ under  $\mathcal{H}_1$ too ($\gamma^{\prime}$ being the probability of the token to be in the green list for a watermarked text and $l$ the size of the text). The parameter $\gamma^{\prime}$  usually cannot be computed theoretically but is simple to estimate empirically. It would have been interesting to provide an estimation of this parameter with and without the selector network to estimate a theoretical gain/loss in detectability. More difficult, but certainly most insightful, a theoretical analysis on the distribution of the $\gamma^{\prime}$ within a text and its impact on detectability would have been welcomed in order to provide conditions on when focusing on certain token distributions while disregarding others is beneficial to detectability.

[1] Fernandez, P., Chaffin, A., Tit, K., Chappelier, V., & Furon, T. (2023, December). Three bricks to consolidate watermarks for large language models. In 2023 IEEE International Workshop on Information Forensics and Security (WIFS) (pp. 1-6). IEEE.

[2] Giboulot, E., & Furon, T. (2024, December). WaterMax: breaking the LLM watermark detectability-robustness-quality trade-off. In NeurIPS 2024-38th Conference on Neural Information Processing Systems (pp. 1-34).

---

> ### Author Rebuttal · Authors · 2025-07-29
>
> **Reply to Reviewer GajM**
>
>  Thank you for your thorough review and for pointing out reference error issue which will be fixed by us. We are deeply grateful for the time and effort you have invested in reviewing our paper. Your valuable questions and insights have significantly contributed to enhancing our work.  To address your queries and concerns, we have prepared detailed responses for your questions and concerns and **we genuinely value the attention you give to our work.** We will explain your concerns point by point and provide you with the additional experiments results concerning using better metrics for the sake of better comparison.
>
> > **Q1:** Could the authors improve their evaluation by focusing on the TPR at a low FPR (at least less 10−5 ) instead of providing many insignificant metrics such as AUC, F1 etc..? The best course of action would be to switch to non-asymptotic p-values if times and resources allow it -- since I understand this would entail changing the loss of the selector network.
>
> We are glad that you have pointed out a better metric for us with the aim of better comparison  between the watermarking methods, we have adopted using p-value method **in our additional experiment**. We used a probability of false-alarm small enough to make out the difference between different methods in terms of detectability instead of metrics such as tpr@1, tpr@2, which are frequently used by previous works, but may not be sufficient enough in our case.
>
>
>
> **Additional Experiments:**
>
> **We genuinely value the opportunity to clarify and address your concerns about  better detection metrics. Taking advantage of the rebuttal period, we have conducted additional experiments to provide a more comprehensive evaluation of our watermarking method and further validate the effectiveness of the proposed method.** We have adopted the ELI5 dataset which is a long form Question Answering dataset, we randomly selected 500 piece of question-answer pairs from the dataset.
>
> We employed the selective watermarking network previously trained on the C4 dataset. Using it is sufficient for the limited time being. In evaluating detectability, we followed your suggestion and adopted the use of p-values under a small false alarm rate, which provides a more rigorous measure of detectability instead of more commonly used metrics in watermarking papers such as FPR@1 and FPR@2 . As for text quality, we used the recall variant of ROUGE-L, a simpler and more computationally efficient metric.
>
> | Method       | LLaMA3.1-8B    |           | OPT-6.7B        |           |
> | ------------ | -------------- | :-------- | --------------- | --------- |
> |              | p-value < 1e-8 | Rouge-L   | p-value < 1e-10 | Rouge-L   |
> | KGW          | 0.984          | 0.230     | 0.994           | 0.244     |
> | Sweet        | 0.986          | 0.221     | **0.996**       | 0.252     |
> | TS-watermark | 0.934          | 0.225     | 0.992           | 0.251     |
> | LTW-1 (ours) | 0.982          | **0.251** | 0.992           | **0.256** |
> | LTW-0 (ours) | **0.988**      | 0.242     | 0.994           | 0.249     |
>
>  Our experimental results in the table demonstrate that our method demonstrate high detectability as well as text quality. Specifically, while Sweet has a performance slightly better than ours in terms of detectability when generated using OPT-6.7b on this task, our methods shows better overall performance, **especially the text quality when generated using LLaMA3.1-8B under this task**. Compared with TS-watermark ,which was trained to adaptively provide gamma and delta hyperparameters using the C4 dataset, **our method shows better generalization ability on a different task**.
>
> We hope that our rebuttal could properly address your concerns and questions.  If so, we would deeply appreciate it if you could kindly raise your score (Rating: 3: Borderline reject), If not, please let us know your further concerns. Thank you for your invaluable time and consideration.
>
> ---
>
>
>
> > **Q2:**  Can you provide a rationale for the choice of LLMs and maybe provide an empirical analysis of the distribution of their entropy for each generation step and for different texts following my "**Uninformed choice of LLM**" comment ?
>
> Thank you for your valuable feedback concerning the choice of LLMs', especially regarding choosing low, medium and high entropy models. In our additional experiment, we have incorporated using Llama3.1-8B together with OPT-6.7B under the task of ELI5, an Long Form Question Answering, we also analysis of the distribution of the entropy of currently used models across the two tasks. As for the reason that we have chosen the OPT model is because it is one of the most frequently used model in the works[1-5] on watermarking ever since the KGW watermark[1].
>
> Below is the **analysis of entropy :**
>
> For the original experiment using OPT and GPT-J on the C4 dataset, they have similar entropy distributions, OPT-6.7B has an average entropy of 1.9 while GPT-j has an average entropy of 2.2.
>
> As for task of  Long Form Question Answering using ELI5 dataset in the additional experiments, the task has a higher entropy. For Llama3.1-8b, it has an average entropy of 2.2 while OPT-6.7B has an average entropy of 2.7.
>
> From the entropy across the two tasks using different models , you can find out that models that have a higher entropy tends to yield better overall detectability,
>
> [1] A Watermark for Large Language Models，ICML 2023
>
> [2] Context-aware Watermark with Semantic Balanced Green-red Lists for Large Language Models. emnlp 2024
>
> [3]  Robust distortion-free watermarks for language models,TMLR
>
> [4]  Token-Specific Watermarking with Enhanced Detectability and Semantic Coherence for Large Language Models,ICML 2024
>
> [5] Adaptive Text Watermark for Large Language Models ,ICML2024
>
> ---
>
>
>
> > **Q3:** Some theoretical analysis about probability of drawing a green token and impact of the selector network on detection.
>
> Thank you for been interesting for for your valuable feedback concerning theoretical analysis about $\\gamma'$ as well as impact on detectability . We would like to provide you with a simple analysis hoping to clarify your worries.
>
> In our method, we like to analysis $\\gamma'$ from two perspectives,the generated text as a whole as well as only those that have been chosen to be watermarked and detected. The parameter $\\gamma$ is the green-list ratio for KGW type watermarks, it is often a fixed hyperparameters throughout the generation and can be used for estimate empirically. For our watermark and other selective watermark, you can consider the $\\gamma'$ became 0 when decided not to add watermark to the current token being generated and the preset $\\gamma$ when chosen to be watermarked from the whole text perspective. The distribution of $\\gamma'$ depends on the selection rule, in our case the output of our model. At this time,it may being you concern about how to compute the distribution of the score for the watermark, but if from the perspective of detection, at detection period we only detect that were chosen and added a bias. Those selected tokens were added bias to according to the green-list ratio $\\gamma'$, thus it also follows $\\mathcal{B}(\\gamma', l)$, except $l$ in our case is the tokens being selected instead of the whole text.
>
> ---
>
>
>
> > **Q4:** *(Curiosity)* Why have you chosen a hard binary mask instead of keeping the soft weights of the selector network? I would guess a soft mask to be more flexible, allowing to tune the bias δ automatically for a given distribution. Have you tried it? If yes, was it worse than the binary mask?
>
> This is a design for robustness. For the soft mask we used during training,  the mask is also a detection weight as in Eq.7, a token with a small mask is also give light detection weight, however, though this brings more flexible may be potentially better in terms of text quality. We believe that, comparatively, using the hard mask can better enhance the robustness of the watermarking scheme under paraphrasing attacks. From a design perspective, it is relatively simpler to categorize the outputs of the selector into two distinct classes—"apply watermark" and "do not apply watermark"—and apply watermarking accordingly. During detection, we only need to verify the outputs that were marked for watermarking, which is a simple approach.
>
> However, if a soft mask, as used during training, were to be adopted at inference time, the detector would likewise need to consider the soft mask values as per-token detection weights. In such a case, tokens with very low mask values would contribute minimally to detection, while the opposite for those with a high mask value. While this offers finer granularity, it introduces more complexity and uncertainty—especially more vulnerable under paraphrasing attacks.
>
> Paraphrasing attacks is hard to change "apply watermark" to "do not apply watermark" than changing its values. In our Eq.9, we penalizing those unmarklike outputs, making most outputs further form the threshold. In contrast, soft mask values are **continuous and more sensitive** to subtle changes, meaning that a paraphrasing attack may cause small shifts in mask values (e.g., a slightly increased value for a previously low-scored token, or a slightly decreased one for a previously high-scored token). These small perturbations can **accumulate** and increase the **uncertainty of watermark detection**, thus reducing robustness.

---

> > ### Comment · Reviewer_GajM · 2025-08-04
> >
> > I thank the reviewers for taking the time to incorporate the changes I asked for.
> >
> > From what I can see, the TPR @ small FPR is still very high for every method. It is once again difficult for me to see any difference between either of the proposed scheme. The only two reasons I can think of is either too high of a temperature of too long of a text (how long is the text by the way?). This is why I would have appreciated to have an average p-value instead of a TPR, this would have allowed a better discrimination between the different schemes. Similarly, the ROUGE score differences do not seem significant to me.
> >
> > For the analysis of $\gamma$, I was just asking the author to compute the actual/empirical $\gamma^{\prime}$ on a sample of texts. This would have actually been a great way to compare schemes (since a higher $\gamma^{\prime}$  leads to lower p-values by definition.
> >
> > For Q4, I understand the choice from the authors. I believe they could have kept a detector which takes a 0-1 decision (does the token belongs to the green-list) while keeping the soft mask **during the embedding** to improve text quality. But maybe I am mistaken.

---

> > > ### Comment · Reviewer_GajM · 2025-08-04
> > > **Current assessement**
> > >
> > > The authors have answered all my questions, not entirely to my satisfaction -- it is still hard to assess the difference of performance between each scheme) but they nevertheless engaged with my concerns. I am fine with a slight increase in score towards a borderline accept.

---

> > > > ### Author Response · Authors · 2025-08-04
> > > >
> > > > Thank you very much for your encouraging and positive feedback on our paper. We are deeply grateful for your consideration that we have addressed some of your concerns. Your thoughtful evaluation and constructive feedback are invaluable to us, and we sincerely appreciate your willingness to raise your from Rating 3 (borderline reject)  to Rating 4 (borderline accept).

---

> > > > ### Author Response · Authors · 2025-08-08
> > > > **Gentle Reminder for Clicking Acknowledgement Button and Updating Current Score**
> > > >
> > > > Dear  Reviewer GajM
> > > >
> > > > I hope this message finds you well. As the rebuttal phase is approaching its end( **Aug 8, 11.59pm AoE**), I just wanted to kindly remind you to **click the Acknowledgement button** once you have finished reviewing this responses.
> > > >
> > > > Also, regarding our earlier discussion where you mentioned that my rebuttal addressed your concerns and you would consider adjusting the score accordingly, I just wanted to gently tell you that this has **not yet been reflected in the system**.
> > > >
> > > > I truly appreciate the time and effort you have devoted to reviewing my work, and I am grateful for your constructive feedback throughout the process.
> > > >
> > > > Best regards

---

### Official Review · Reviewer_Ezdj · 2025-07-03

**Clarity:** 3
**Significance:** 3
**Originality:** 3
**Rating:** 4
**Confidence:** 4

**Summary:**

This paper introduces "Learning to Watermark" (LTW), a novel selective watermarking framework for Large Language Models (LLMs) that leverages multi-objective optimization to balance watermark detectability and text quality. Traditional watermarking methods often face a trade-off between these two objectives, leading to degraded text quality or reduced detectability. LTW addresses this by employing a lightweight neural network, the Selector Network, which adaptively decides when to apply watermarks based on sentence embeddings, token entropy, and the current watermarking ratio. The network is trained using two specifically designed loss functions that guide it toward Pareto-optimal solutions, ensuring high detectability without compromising text quality.

The authors evaluate LTW by integrating it with two baseline watermarking methods, KGW and Unigram, and demonstrate significant improvements in text quality while maintaining or even enhancing detectability. Experimental results show that LTW achieves lower perplexity and higher cosine similarity with reference texts compared to existing methods, indicating better semantic coherence and text quality. Additionally, LTW exhibits superior robustness against paraphrase attacks, making it a promising approach for practical LLM watermarking applications. The paper contributes a new perspective on selective watermarking strategies, emphasizing the importance of adaptive decision-making in maintaining both security and usability.

**Questions:**

1. The current model primarily controls text quality and watermarking capability through loss functions. Does the weight of different losses have an impact on the model's performance? Additionally, does the number of tokens selected or the window size used in the model affect its effectiveness?

2. How does the model ensure robustness against paraphrasing or deletion attacks? It seems that the method does not explicitly consider these types of attacks, which could be a potential vulnerability.

**Ethical Concerns:**

["NO or VERY MINOR ethics concerns only"]

**Final Justification:**

4: Borderline accept

**Limitations:**

yes

**Paper Formatting Concerns:**

no formatting issues in this paper

**Quality:**

3

**Strengths And Weaknesses:**

## Strengths

- The paper introduces a novel selective watermarking framework, LTW, which leverages multi-objective optimization to balance watermark detectability and text quality. This approach is innovative and addresses a significant challenge in the field of LLM watermarking.

- The authors employ the Multiple Gradient Descent Algorithm (MGDA) to solve the multi-objective optimization problem, ensuring that the framework achieves Pareto-optimal solutions. This method effectively harmonizes the competing goals of high detectability and high text quality.

- The paper provides extensive experimental evaluations across multiple models and datasets, demonstrating that LTW significantly enhances text quality without compromising detectability. The results are compelling and show that LTW outperforms existing watermarking methods in both text quality and robustness.

## Weaknesses

- The paper only evaluates the watermarking method against a single type of attack, specifically the Dipper Attack. It lacks experiments on other types of attacks, such as scrubbing and spoofing, which are important for a comprehensive understanding of the method's robustness.

- The experiments are limited to one dataset, which may not be sufficient to demonstrate the generalizability of the proposed method across different datasets and domains.

- In Table 3, as the threshold increases, the effect of the selector appears to diminish. The paper does not include ablation studies on other components of the model, making it difficult to assess the individual contributions of each module to the overall performance.

---

> ### Author Rebuttal · Authors · 2025-07-29
>
> **Reply to Reviewer Ezdj**
>
> We thank the reviewer for the insightful and valuable comments. We respond to your comment and problems as follows and sincerely hope that our rebuttal could properly address your concerns.
>
> > **Q1:** The experiments are limited to one dataset, which may not be sufficient to demonstrate the generalizability of the proposed method across different datasets and domains.
>
> **Reply:** **We did an additional experiment using the ELI5 dataset to demonstrate the generalizability of the proposed method across different datasets.**
>
>  The ELI5 dataset is a long form Question Answering dataset, we randomly selected 500 piece of question-answer pairs from the dataset and did the extra experiment on it.
>
> We employed the selective watermarking network previously trained on the C4 dataset, without training it on the new task and dataset. In evaluating detectability, we followed suggestion of reviewer GajM and adopted the use of p-values under a small false alarm rate, which provides a more rigorous measure of detectability instead of more commonly used metrics in watermarking papers such as FPR@1 and FPR@2 . As for text quality, we used the recall variant of ROUGE-L, a simpler and more computationally efficient metric.
>
> | Method       | LLaMA3.1-8B    |           | OPT-6.7B        |           |
> | ------------ | -------------- | :-------- | --------------- | --------- |
> |              | p-value < 1e-8 | Rouge-L   | p-value < 1e-10 | Rouge-L   |
> | KGW          | 0.984          | 0.230     | 0.994           | 0.244     |
> | Sweet        | 0.986          | 0.221     | **0.996**       | 0.252     |
> | TS-watermark | 0.934          | 0.225     | 0.992           | 0.251     |
> | LTW-1 (ours) | 0.982          | **0.251** | 0.992           | **0.256** |
> | LTW-0 (ours) | **0.988**      | 0.242     | 0.994           | 0.249     |
>
>  Our experimental results in the table demonstrate that our method demonstrate high detectability as well as text quality. Specifically, while Sweet has a performance slightly better than ours in terms of detectability when generated using OPT-6.7b on this task, our methods shows better overall performance, **especially the text quality when generated using LLaMA3.1-8B under this task**. **Compared with TS-watermark ,which was trained to adaptively provide gamma and delta hyperparameters using the C4 dataset**, **our method shows better generalization ability on a different task**.
>
> ---
>
> >  **Q2:** Does the weight of different losses have an impact on the model's performance? Additionally, does the number of tokens selected or the window size used in the model affect its effectiveness?
>
> **Question about Weight of different Losses:**
>
> **Reply:** The weights of these loss functions indeed play a important role in our method. Through extensive experiments, we have developed some empirical guidelines.
>
> First, for the loss term in Eq. 9, its weight should not be set too high. We typically set it in the range of 0.1–0.15, as its main function is to encourage the output to be more mask-like. This not only helps during training but also contributes to the robustness of the watermarking method—a point we will elaborate on later. Second, the weight corresponding to Eq. 7 should neither be too large nor too small. Improper weighting can lead to unreasonable watermarking ratios (either too high or too low), which directly affects both utility and detectability. The other loss terms are relatively more flexible and can be adjusted with less sensitivity to overall performance.
>
> **Question about Window Size:**
>
> **Reply:** We deliberately chose a **moderate window size**—neither too small nor too large—for the sentence embedding used by the selector network. A window size that is **too small** may cause the sentence embedding to degrade like those of a token-level representation, making it less robust under paraphrasing attacks. This is because paraphrasing typically preserves the overall sentence meaning while altering individual words through synonym substitution or structural changes. At the token level, such variations can affect the selector network’s output, making it less robust to these attacks. In contrast, by selecting an appropriately sized window, we can maintain semantic consistency with the paraphrased version, thus improving the **robustness** of the selector. We will further elaborate later.
>
> On the other hand, if the window size is **too large**, each new token generation results in only a minor change to the window—dropping the oldest token and adding the most recent one. From a semantic perspective, this leads to minimal variation between consecutive windows, which may reduce the **sensitivity** of the selector. As a consequence, both during training and inference, the selector may fail to be of use to the token currently being generated.
>
> Therefore, choosing a **moderate and well-balanced window size** is crucial to ensure both robustness to paraphrasing and effectiveness in guiding selective watermarking.
>
> ---
>
> > **Q3:** How does the model ensure robustness against paraphrasing or deletion attacks? It seems that the method does not explicitly consider these types of attacks, which could be a potential vulnerability.
>
>
> **Reply:** We appreciate your concerns and questions about robustness which we didn't explain clearly in the paper due to page limitations. We genuinely value the opportunity to clarify and address your concerns about robustness. We will **explain** **the measures during watermark design to prevent ill impact on robustness step by step** :
>
> **Measures to ensure Robustness**:
>
>  Reviewer GajM asked us out of *Curiosity* that why have we chosen a hard binary mask instead of keeping the soft weights of the selector network like during training, which can be more flexible. This is a design for robustness. For the soft mask used during training, the mask is also a detection weight as in Eq.7, a token with a small mask is also give light detection weight, however, though this brings more flexible may be potentially better in terms of text quality. We believe that, comparatively, using the hard mask can better enhance the robustness of the watermarking scheme under paraphrasing attacks. From a design perspective, it is relatively simpler to categorize the outputs of the selector into two distinct classes—"apply watermark" and "do not apply watermark"—and apply watermarking accordingly. During detection, we only need to verify the outputs that were marked for watermarking, which is a simple approach.
>
> However, if a soft mask, as used during training, were to be adopted at inference time, the detector would likewise need to consider the soft mask values as per-token detection weights. In such a case, tokens with very low mask values would contribute minimally to detection, while the opposite for those with a high mask value. While this offers finer granularity, it introduces more complexity and uncertainty—especially more vulnerable under paraphrasing attacks.
>
> The inputs of our selector includes a suitable window size k for obtaining the sentence embedding, using a k that is too short or if simply using the previous token embedding which would be easier to obtain and no need for the extra sentence would make it vulnerable to paraphrasing attacks since the previous may change due to the attack. Previous work[1] have shown that sentence semantic meanings are robust under paraphrasing attacks. (In their work ,they replaced KGW's[2] hashing method with a semantic one, improving robustness). This is a method done to enhance the robustness of our network output.
>
> Paraphrasing attacks is hard to change "apply watermark" to "do not apply watermark" than changing its values because in our Eq.9, we penalizing those unmarklike outputs, making most outputs further form the threshold. In contrast, soft mask values are **continuous and more sensitive** to subtle changes, meaning that a paraphrasing attack may cause small shifts in mask values (e.g., a slightly increased value for a previously low-scored token, or a slightly decreased one for a previously high-scored token). These small perturbations can **accumulate** and increase the **uncertainty of watermark detection**, thus reducing robustness.
>
> These are some of the measures taken during watermark design to ensure robustness against paraphrasing. In addition we aim to provide a selective watermarking method which can be adopted to different watermarking method. For example, the work of SIR[1] can be combined with ours by changing the hashing method from the previous tokens (like the case of KGW) to using a semantic based one for further improving robustness under paraphrasing attacks.
>
> [1] A Semantic Invariant Robust Watermark for Large Language Models，ICLR2024
>
> [2] A Watermark for Large Language Models，ICML 2023

---

> ### Comment · Reviewer_Ezdj · 2025-08-05
>
> Thank you for your detailed rebuttal and for conducting additional experiments to address my concerns. I have carefully reviewed your responses and the new results. Please find my feedback below.
>
> **Regarding Q1: Generalizability**
>
> I appreciate the new experiment on the ELI5 dataset. This is a valuable step towards demonstrating the generalizability of your method.
>
> However, while your method achieves high p-values, the performance of the baseline methods is also exceptionally high. For instance, on LLaMA3.1-8B, KGW (0.984) and Sweet (0.986) show detection rates that are highly competitive with your LTW-1 (0.982) and LTW-0 (0.988). The performance margins are not significant enough to clearly establish the superiority of your method in this new, out-of-domain setting. A more compelling case would require demonstrating a more substantial advantage over these strong baselines.
>
> **Regarding Q2: Impact of Loss Weights and Window Size**
>
> Thank you for the detailed explanations regarding the empirical guidelines for setting the loss weights and choosing the window size. The intuitions you provided are insightful.
>
> However, this discussion remains largely qualitative. To substantiate these claims, the paper would be significantly strengthened by quantitative evidence. I would expect to see ablation studies that empirically validate these choices. For example:
> *   A table or plot illustrating how key metrics (e.g., p-value, Rouge-L) are affected by varying the weights of the loss terms (especially for Eq. 7 and Eq. 9).
> *   An analysis showing the model's performance with different window sizes (e.g., smaller, moderate, and larger) to support your claim that a moderate size is optimal.
>
> Without such experiments, these crucial design choices lack the necessary empirical validation.
>
> **Regarding Weakness 3: Lack of Ablation Studies for Model Components**
>
> Furthermore, I noticed that my original concern about the lack of ablation studies for the model's components (which I believe was noted as Weakness 3) has not been addressed in the rebuttal.
>
> A comprehensive ablation analysis is essential to understand the contribution of each part of your proposed architecture. For instance, what is the specific impact of the selector network? How does the system perform with a simpler or non-learned selection mechanism? Such an analysis would justify the complexity of your model and provide deeper insights into why it works. I strongly encourage you to include these ablation studies in your revision.
>
> Overall, I maintain my current score.

---

### Note · Authors · 2025-08-13

# Deepest thanks for your relentless dedication!

Dear NeurlPS 2025 PC,SAC and AC, we thank you for the time and effort you have dedicated to. We thank all reviewers for their constructive feedback. Below we consolidate and clarify our advantages and the key points raised across the four official reviewers (Reviewer Ezdj, GajM,v5TR, 9Hqn) and provide final clarifications for AC consideration.



1. **Our Advantages**:
   - **Novel method:** We proposed a novel way of selectively watermarking LLMs. We utilized a trained network as well as several  informative factors for the network to decide whether to apply watermark.
   - **Extensibility:** Rather than solely a watermarking method, we provide a novel watermarking framework that can adopt other baseline watermarking method to use our selective watermarking approach.
   - **Better Detectability and Quality:** Our method achieves better text quality while still having similarity or a better performance of detectability.
   - **Moreover:** Our method is innovative and have compelling results (Reviewer Ezdj), Clear and well-written (Reviewer GajM,v5TR) with Nice analysis (Reviewer GajM), providing an optimized method and extensive experiments(Reviewer 9Hqn)
2. **Rebuttals to criticisms:**
   - We have done additional experiments using Long Form Question Answering dataset ,using more recent LLMs and other metrics and addressed the reviewers' concerns about  the generalization ability of the method to other tasks and models.
   - We have explained the designs of our method on robustness as well as why we didn't adopt the  the soft weights of the selector network used during training which we didn't explain previously due to page limitations in the main text. Our explanation during rebuttal addressing the reviewers concerns about it.

---

### Decision · Program_Chairs · 2025-09-17

**Decision:**

Accept (poster)

**Comment:**

This paper received all positive reviews. After discussion, reviewers agreed to accept this paper.  The authors are encouraged to address the major concerns in the camera-ready version.
1. Add experimental evaluation on other types of attacks
2. Justify the choice of evaluation metrics
3. Add more ablation studies.

The authors’ rebuttal and messages were carefully ready, discussed, and considered.